# VSIG4 inhibits proinflammatory macrophage activation by reprogramming mitochondrial pyruvate metabolism

Jialin Li[1], Bo Diao[1], Sheng Guo[1], Xiaoyong Huang[1], Chengying Yang[1], Zeqing Feng[1], Weiming Yan[2], Qin Ning[2], Lixin Zheng[3], Yongwen Chen[1] & Yuzhang Wu[1]

Exacerbation of macrophage-mediated inflammation contributes to pathogenesis of various inflammatory diseases, but the immunometabolic programs underlying regulation of macrophage activation are unclear. Here we show that V-set immunoglobulin-domain-containing 4 (VSIG4), a B7 family-related protein that is expressed by resting macrophages, inhibits macrophage activation in response to lipopolysaccharide. $Vsig4^{-/-}$ mice are susceptible to high-fat diet-caused obesity and murine hepatitis virus strain-3 (MHV-3)-induced fulminant hepatitis due to excessive macrophage-dependent inflammation. VSIG4 activates the PI3K/ Akt–STAT3 pathway, leading to pyruvate dehydrogenase kinase-2 (PDK2) upregulation and subsequent phosphorylation of pyruvate dehydrogenase, which results in reduction in pyruvate/acetyl-CoA conversion, mitochondrial reactive oxygen species secretion, and macrophage inhibition. Conversely, interruption of $Vsig4$ or $Pdk2$ promotes inflammation. Forced expression of $Vsig4$ in mice ameliorates MHV-3-induced viral fulminant hepatitis. These data show that VSIG4 negatively regulates macrophage activation by reprogramming mitochondrial pyruvate metabolism.

[1] Institute of Immunology, PLA, Third Military Medical University, Chongqing, 400038, China. [2] Institute of Infectious Disease, Tongji Hospital of Tongji Medical College, Huazhong University of Science and Technology, Wuhan, 430030, China. [3] Laboratory of Immunology, National Institute of Allergy and Infectious Diseases, NIH, Bethesda, Maryland, MD 20892, USA. Jialin Li and Bo Diao contributed equally to this work. Correspondence and requests for materials should be addressed to Y.C. (email: yongwench@163.com) or to Y.W. (email: wuyuzhang@tmmu.edu.cn)

Macrophages are essential for innate immunity owing to functions in host defense, tissue development and homeostasis[1]. Macrophage functional disparity is attributed to two distinct subgroups, namely M1 (classically activated) and M2 (alternatively activated) macrophages[2]. M1 macrophages have been implicated in initiating and sustaining inflammation in response to INF-γ and/or lipopolysaccharide (LPS), whereas IL-4 or IL-13 polarized M2 macrophages seem to have immunoregulatory functions in parasitic infections, tissue inflammation, remodeling of damaged tissue, and tumor progression[3]. Uncontrolled M1 activation can cause tissue damage and pathogenesis in inflammatory diseases, including atherosclerosis, obesity, diabetes, rheumatoid arthritis and hepatitis[4]. In a model of mouse obesity caused by high-fat diet (HFD), the accumulation of adipose tissue macrophages (ATMs) within the white adipose tissue (WAT) involves remodeling of the enlarged WAT and induction of insulin resistance via secreting proinflammatory mediators IL-1β and TNF[5, 6]. Moreover, in a viral fulminant hepatitis model, murine hepatitis virus strain-3 (MHV-3) infection induces a macrophage-dependent cytokine storm of IL-1, TNF, TGF-β, leukotriene B4, and pro-coagulant fibrinogen-like protein-2 (FGL2), which causes fibrin deposition in the liver and results in acute hepatic necrosis and lethality of susceptible mice[7]. Therefore, attenuation of macrophage-mediated inflammation is a plausible strategy for treating inflammatory disorders.

The regulation of macrophage activation has been extensively studied, with evidence suggesting the involvement of multiple intracellular signaling regulators, including membrane molecules, small noncoding RNAs, microRNAs and epigenetic-associated mechanisms[8]. Evidence from metabolic screening and microarray analyses shows that LPS-activated macrophages have alterations in mitochondrial metabolites, indicating that reprogramming of mitochondrial metabolism may be involved in the regulation of macrophage activation[9, 10]. For example, $NAD^+$, as a mitochondrial intermediate metabolite, can inhibit inflammation through inactivating transcription factor NF-κB[11, 12]. Conversely, succinate, another important mitochondrial metabolite, accumulates in LPS-activated macrophages and promotes the transcription of Il1b by stabilizing hypoxia-inducible factor 1α (HIF-1α)[13]. Moreover, mitochondrial reactive oxygen species (mtROS) can activate the NLRP3 inflammasome and trigger bioprocessing of proinflammatory cytokines including pro-IL-1β and pro-IL-18 in macrophages[14, 15]. Therefore, inhibition of autophagy, which impairs the removal of ROS-generating mitochondria and causes NLRP3 inflammasome activation and IL-1β secretion, attenuates glucose tolerance and insulin sensitivity[16]. Additionally, mtROS can induce M1 activation through activating NF-κB and stabilizing HIF-1α[17, 18]. Therefore, suppression of mtROS secretion can mitigate pathogenesis in alcoholic steatohepatitis and reduce lethality in endotoxin-mediated fulminant hepatitis[19]. Nevertheless, the mechanisms underlying reprogramming of mitochondrial metabolism during macrophage activation are unclear.

V-set immunoglobulin-domain-containing 4 (VSIG4) is a membrane protein belonging to complement receptor of the immunoglobulin superfamily (CRIg)[20, 21]. By binding complement component C3b, VSIG4 mediates clearance of C3b-opsonized pathogens, such as Listeria monocytogenes and Staphylococcus aureus[21]. The expression of VSIG4 is restricted to tissue macrophages, including peritoneal macrophages and liver-residential Kupffer cells. Moreover, VSIG4 marks a subset of macrophages that associates with diabetes resistance[22]. VSIG4 can functionally inhibit IL-2 production and T-cell proliferation by binding an unidentified T-cell ligand or receptor[20]. Interestingly, a VSIG4-Fc fusion protein seems to protect against development of experimental arthritis[23], experimental autoimmune uveoretinitis[24], and immune-mediated liver injuries[25], suggesting

that VSIG4 can deliver anti-inflammatory signals. Here, we show that VSIG4 antagonizes activation signals in macrophages by stimulating PI3K/Akt–STAT3 cascades, augmenting expression of pyruvate dehydrogenase kinase-2 (PDK2), and inhibiting pyruvate dehydrogenase (PDH) activity via phosphorylation. Therefore, VSIG4 restricts pyruvate metabolism in the mitochondria during oxidative phosphorylation (OXPHOS), resulting in suppression of mtROS secretion and M1 differentiation. Conversely, Vsig4 or Pdk2 deficiency enhances macrophage activation. $Vsig4^{-/-}$ mice are more susceptible to HFD-induced obesity in association with insulin resistance. These mice manifest with more severe liver damage and mortality as a result of MHV-3 infection, probably owing to overloading of macrophage-mediated inflammation in vivo. Interestingly, forced over-expression of Vsig4 ameliorates MHV-3-induced viral fulminant hepatitis. These data identify an inhibitory function of VSIG4 in macrophage-mediated inflammation.

## Results

**$Vsig4^{-/-}$ mice are more susceptible to HFD-induced obesity.** Macrophages actively contribute to the pathogenesis of diet-induced obesity[5, 6], making them seemingly a good model for examination of the biological functions of VSIG4. For this, age-matched C57BL/6 wild type (WT) and $Vsig4^{-/-}$ mice were fed a HFD. Interestingly, we found that $Vsig4^{-/-}$ mice gained significantly more body masses than their WT controls after 5 weeks of HFD feeding (Fig. 1a). However, no significant differences of body masses between the two groups fed a normal chow diet (NCD) (Supplementary Fig. 1a). Analytical microCT (μCT) imaging revealed an increase in fat mass throughout the bodies of the $Vsig4^{-/-}$ obese mice, with substantially more abundant visceral fat (Fig. 1b). $Vsig4^{-/-}$ obese mice also manifested with significant increases in abdominal wall fat and perirenal fat compared to the WT controls. In parallel, we also observed a substantial increase in the serum levels of triglyceride, cholesterol and free fatty acid, which correlated to increases in body fat in $Vsig4^{-/-}$ obese mice (Fig. 1c). Furthermore, $Vsig4^{-/-}$ obese mice tended to develop with high liver triglyceride levels and steatosis (Fig. 1d), as well as enlarge adipocytes (Fig. 1e). These data imply that Vsig4 deficiency renders mice susceptible to HFD-induced obesity.

Obesity is often associated with insulin resistance[5], a fact that led us to postulate that $Vsig4^{-/-}$ mice might have a disturbed glucose metabolism. We next examined the blood glucose levels of these obese mice. $Vsig4^{-/-}$ obese mice exhibited statistically significant elevations in blood glucose relative to WT littermates after 15 h of fasting. Moreover, the 5-h-fasting insulin levels were also substantially elevated (Fig. 1f). An oral glucose tolerance test (GTT) illustrated that $Vsig4^{-/-}$ obese mice had significantly higher serum levels of glucose and insulin in response to the glucose load compared to their WT littermate controls, suggesting a severely impaired glucose metabolism (Fig. 1g). Nevertheless, the blood glucose metabolism was similar between the two groups under NCD conditions (Supplementary Fig. 1b, c). The insulin tolerance test (ITT) also indicated the existence of a significantly more resistance in $Vsig4^{-/-}$ obese mice compared to the WT controls (Fig. 1g). Additionally, as compared to WT controls, western blot analyses revealed that $Vsig4^{-/-}$ obese mice had diminished phosphorylation of IRS-1 (p-IRS-1) and phosphorylation of Akt (p-$Akt^{ser473}$) in the visceral adipose tissue (VAT), muscle and liver tissues after administrated with insulin (Fig. 1h). These data indicate that Vsig4 deficiency promotes obesity-associated insulin resistance.

To better understand the mechanisms by which the VSIG4 pathway prevents weight gain during HFD consumption, we

addressed the food intake of these mice. There were no obvious changes in food intake and stool output between the WT and *Vsig4*−/− mice under HFD conditions (Supplementary Fig. 2), implying that the differences in weight gain are due to reasons other than food consumption. We therefore examined the status of the ATMs. Compared to WT littermates, flow cytometric assay showed that the ATMs from *Vsig4*−/− obese mice expressed higher amounts of proinflammatory factors (like pro-IL-1β, IFN-γ, and TNF), which have been described to actively participate into the pathogenesis of HFD-caused obesity and insulin resistance[26, 27] (Fig. 1i). In parallel, dramatically higher levels of these cytokines were accumulated in the VAT of *Vsig4*−/− mice, as detected by qRT-PCR (Fig. 1j) and western blot (Fig. 1k), respectively. These data suggest that *Vsig4* deficiency initiates macrophage-mediated inflammation, which triggers HFD-induced obesity and insulin resistance.

**Vsig4−/− mice exacerbate MHV-3-induced fulminant hepatitis.** We further investigated the potential role of *Vsig4* in viral ful-minant hepatitis caused by MHV-3 infection, in which the virus-induced exaggerated inflammation causes severe pathogenesis largely due to the M1 macrophage-dependent "cytokine storm"[7]. *Vsig4*−/− and congenic C57BL/6 WT littermates were infected with MHV-3 (100 PFU/mouse). Noticeably, *Vsig4*−/− mice died rapidly following MHV-3 infection in contrast to WT littermates (log-rank test, $p = 0.0339$, Fig. 2a). H&E staining showed that the infected *Vsig4*−/− animals had more severe liver necrosis and hepatocyte apoptosis at 48 h and 72 h of infection (Fig. 2b), along with significantly higher levels of serum alanine aminotransferase (ALT) and aspartate aminotransferase (AST), the two liver damage indicating enzymes released into the blood compared with WT controls (Fig. 2c). Plaque assay data also showed that *Vsig4* deficiency promoted virus replication in liver tissues

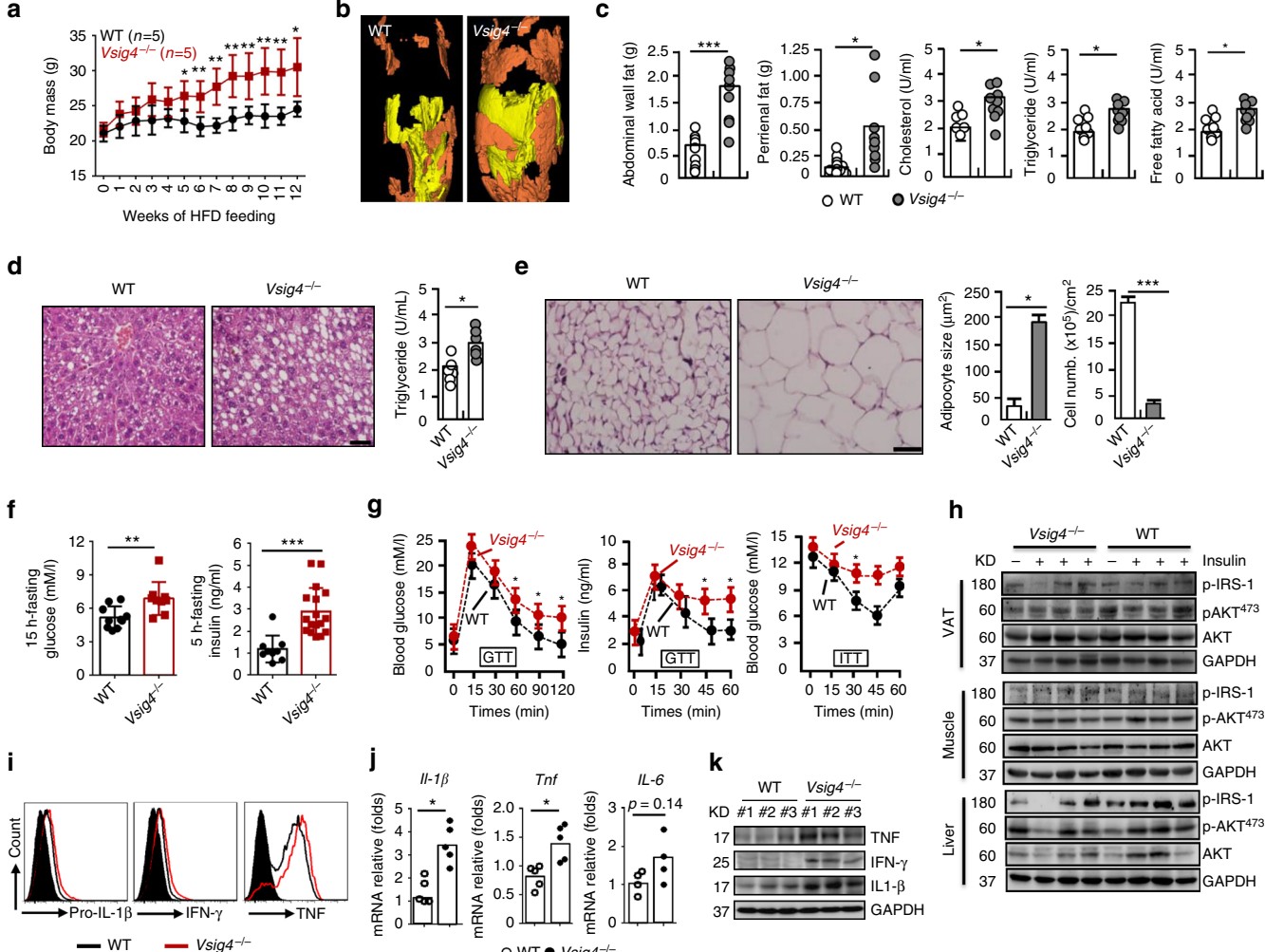

**Fig. 1** *Vsig4*−/− mice are more susceptible to HFD-induced obesity with insulin resistance. Eight-week-old male *Vsig4*−/− mice and age-matched C57BL/6 WT controls were fed a HFD. **a** Body weight was measured and compared. The obese mice were sacrificed after 10 weeks of HFD feeding. **b** Fat distribution was detected by μCT. Yellow indicates subcutaneous fat and brown indicates that visceral fat. **c** Measurement of abdominal wall fat, perirenal fat, serum triglyceride, cholesterol, and free fatty acid. **d** Representative liver H&E staining (left), and intrahepatic triglyceride contents (right), scale bar = 20 μm, n = 10 per group. **e** Representative the architecture of adipose tissues stained by H&E (left), adipocyte size and cell numbers was calculated (right), scale bar = 20 μm, n = 10 per group. **f** The 15-h-fasting blood glucose levels and 5-h-fasting serum insulin levels. **g** GTT and ITT were performed in theses obese mice, n = 6 per group. **h** Western blot of the AKT, p-Akt[ser473], and p-IRS-1 in VAT, muscle and liver tissues of obese mice after 4 min of insulin administration, n = 4 per group. ATMs were isolated from obese mice. **i** Flow cytometry analyzing pro-IL-1β, IFN-γ, and TNF. Cytokines in VAT were detected by **j** qRT-PCR and **k** western blot. Error bar, s.e.m. *$p < 0.05$, **$p < 0.01$ and ***$p < 0.001$ (Student's t-test). Data are representative of five (**a**) and three (**c–i**) independent experiments

(Fig. 2d). This suggests that VSIG4 is capable of attenuating MHV-3-induced pathogenesis.

The macrophage-derived FGL2 and proinflammatory cytokines including TNF, IL-1 and IL-6, play essential roles in the pathogenesis of MHV-3-induced fulminant hepatitis[28–30]. We therefore examined the expression of these mediators in peritoneal exudate macrophages (PEMs) and liver tissues. Although *Vsig4* deficiency did not seem to affect the basal expression level of these factors before viral infection (Supplementary Fig. 3), qRT-PCR data showed that MHV-3 infection super-induced *Vsig4*−/− PEMs to express *Fgl2*, *Tnf*, *Il-1β*, and *Il-6* (Fig. 2e). Flow cytometric data also confirmed at protein levels that inflammatory cytokines like pro-IL1-β, TNF, and IL-6, were dramatically increased in virus-infected *Vsig4*−/− PEMs (Fig. 2f), suggesting *Vsig4* deficiency promotes macrophage-derived inflammation in vivo. Consistent with this, *Vsig4* deficiency in liver Kupffer cells also resulted in higher levels of these factors deposited in the infected liver tissues, as detected by qRT-PCR

(Fig. 2e), western blot (Fig. 2g), and immunohistochemistry (Supplementary Fig. 4). Therefore, dramatically higher levels of these proinflammatory cytokines were accumulated in the virus-infected *Vsig4*−/− serum (Fig. 2h). Finally, *Vsig4*−/− mice responded with severe fibrinogen formation, leading to increased liver coagulation and necrosis post infection (Supplementary Fig. 4). VSIG4 has been described to be a negative regulator of T-cell activation[20], nevertheless, the secretion of proinflammatory cytokines (TNF and IFN-γ), and the expression of activated markers (CD25 and CD69) from CD4+ as well as CD8+T cells in 72 h of MHV-3-infected *Vsig4*−/− livers was similar to their WT controls (Supplementary Fig. 5). These results clearly demonstrate that *Vsig4* deficiency exacerbates macrophage-mediated inflammation, which deteriorates MHV-3 virus-induced FH.

**VSIG4 attenuates LPS-induced macrophage activation in vitro.** Our above data from in vivo experiments demonstrated that *Vsig4* deficiency promotes macrophage-derived inflammation, we

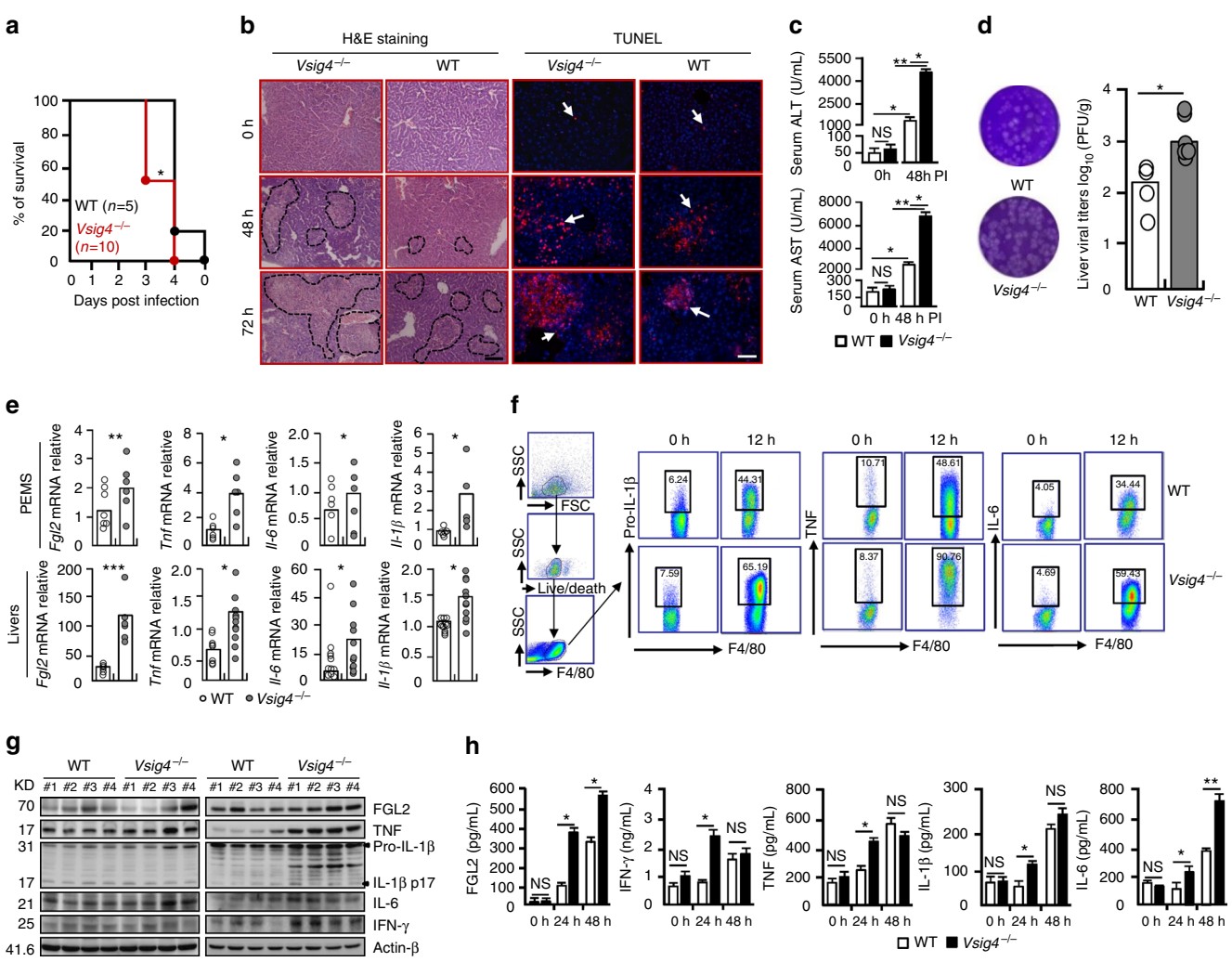

**Fig. 2** *Vsig4* deficiency exacerbates MHV-3-induced fulminant hepatitis. The *Vsig4*−/− mice and age-matched C57BL/6 WT littermates were infected with MHV-3 (100 PFU/mouse) via i.p. injection. **a** The survival was monitored. **b** H&E staining of liver, and TUNEL staining of cell apoptosis, scale bar = 20 μm, n = 5–8 per group, arrow indicated positive cells. **c** Serum ALT and AST levels at 0 h and 48 h post infection (PI), n = 5–8 per group. **d** Plaque assay of virus titers in livers at 48 h PI. **e** qRT-PCR analyzing proinflammatory cytokines in PEMs at 12 h and in liver tissues at 72 h of MHV-3 infection. **f** Flow cytometry analyzing TNF, pro-IL1-β, and IL-6 from PEMs after 12 h of virus infection. **g** Western blot analyzing proinflammatory cytokines in infected livers at 24 h and 48 h PI, n = 4 per group. **h** ELISA of serum concentration of proinflammatory mediators, n = 5–10 per group. Error bar, s.e.m. *p < 0.05, **p < 0.01, ***p < 0.001 and NS, p > 0.05. **a** was analyzed by log-rank test and others are calculated by Student's t-test. Data are representative of six (**a**) and three (**b**–**f**, **h**) independent experiments

then investigated the role of VSIG4 in regulating macrophage activation in response to LPS in vitro. PEMs that are isolated from Vsig4[−/−] mice appeared to present with an abrupt surge of M1-like proinflammatory gene transcripts, such as Il-1β and Tnf, at the very early stage (1 h and 6 h) of LPS exposure (Fig. 3a). This was validated by the existence of higher levels of IL-1β, TNF, and IL-6 protein in the supernatants of LPS-stimulated Vsig4[−/−] PEM cultures compared to the WT counterparts (Fig. 3b). Western blot also confirmed these results (Fig. 3c). In addition, surface expression of M1 activation markers, including B7-H1, B7-DC, B7-H3, and CD40, was higher in LPS-treated Vsig4[−/−] PEMs (Fig. 3d). These results suggest that VSIG4 is important for controlling macrophage activation.

To avoid the cellular heterogeneity of conventional PEMs, we next chose a macrophage line, RAW264.7 cells, as a homogeneous model to examine the functional specificity of VSIG4. RAW264.7 cells are lack of Vsig4 transcription, but they with lentiviral-mediated restoration of Vsig4 expression (Len-Vsig4) exhibited a reduction in LPS-induced M1 gene (Il1b, Il6, and Tnf) transcripts compared to the control counterparts (Fig. 3e). Similarly, ELISA data also showed that the levels of TNF, IL-6, IL-1β, and IL-12p40 protein in the supernatants of the LPS-treated VSIG4[+]RAW264.7 cultures were significantly lower than the controls (Fig. 3f). These VSIG4[+]RAW264.7 cells were also incapable of mounting LPS-induced CD40 upregulation (Fig. 3g). These combined data imply that VSIG4 inhibits LPS-induced macrophage activation in vitro.

Previous work has shown that the complement C3b is the natural ligand of VSIG4[21]. To address whether VSIG4 regulates macrophage activation is dependent on C3b, we used lentiviral vectors to overexpress VSIG4 in C3[−/−] BMDMs and subjected these cells to LPS treatment (2 μg/ml). ELISA showed that the secretion of IL-6 and IL-1β was still dramatically downregulated in VSIG4[+]C3[−/−] BMDMs than their controls (Fig. 3h). These combined data suggest that VSIG4-mediated cytokine production in macrophages is C3b independent.

**VSIG4 reprograms pyruvate metabolism and mtROS generation.** To investigate the molecular mechanisms for VSIG4-mediated macrophage activation, we constructed a platform using VSIG4[+]RAW264.7 cells to test the dependence of macrophage activation on VSIG4. Inspired by recent studies showing that cell metabolism has an important function in supporting macrophage activation and polarization[31], we measured cell metabolism thereafter. Figure 4a shows that VSIG4 did not affect LPS-induced glucose uptake. However, it inhibited lactate, pyruvate and acetyl-CoA accumulation after 6 h of LPS administration. The mitochondrial oxygen consumption rate (OCR) in the macrophages was then investigated, although VSIG4 did not affect oxygen consumption under normal conditions (Supplementary Fig. 6a, b), it appeared to drastically downregulate oxygen consumption after LPS exposure, both in basal and

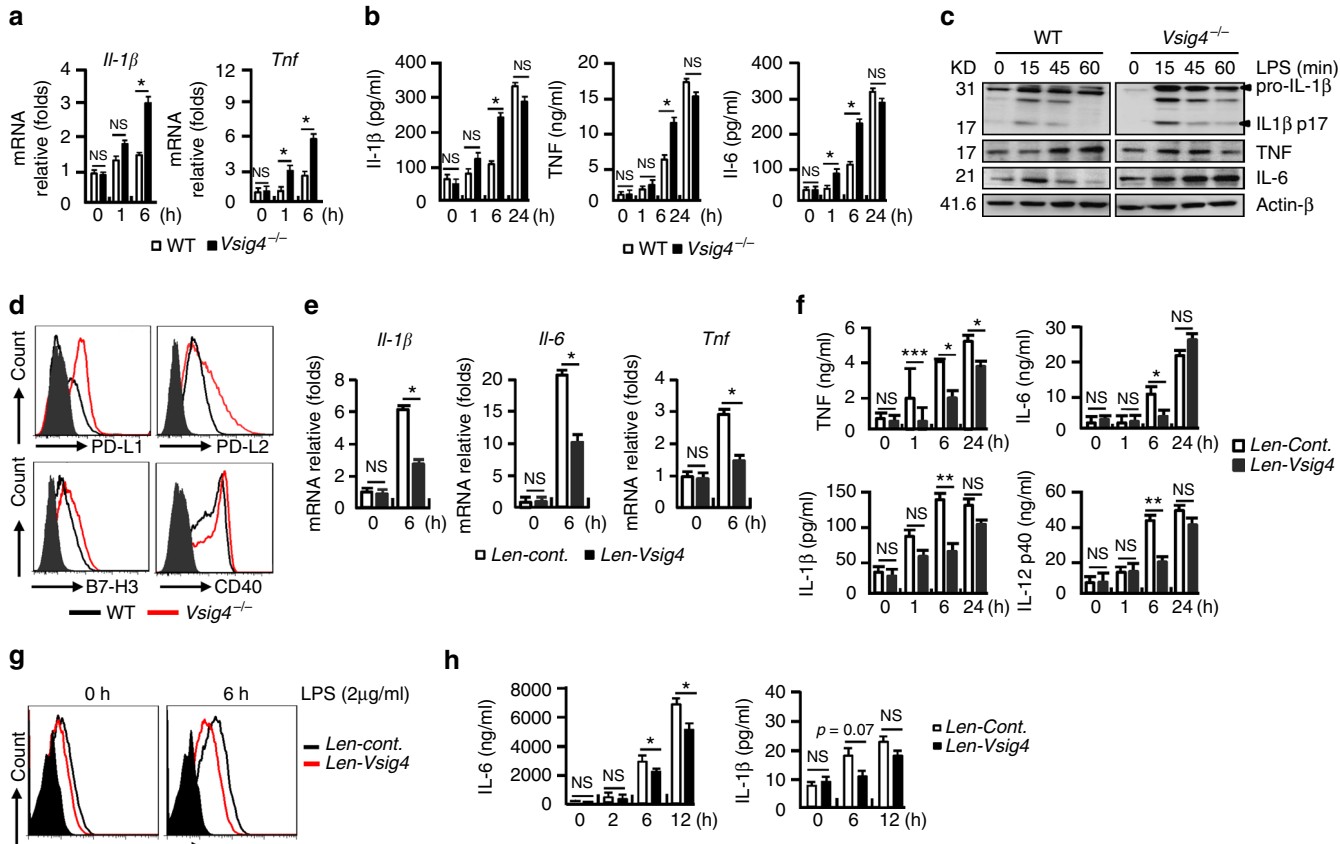

**Fig. 3** VSIG4 impedes LPS-induced macrophage M1 polarization in vitro. PEMs were treated with LPS (2 μg/ml), **a** qRT-PCR analysis of Il-1β and Tnf transcripts. **b** ELISA of cytokines in cultured supernatants. **c** Western blot analyzing cytokine protein expression. **d** Flow cytometry analyzing surface expression of activation markers. RAW264.7 cells stably infected with lentiviral control vectors (Len-cont.) or vectors encoding Vsig4 (Len-Vsig4), cells were further treated with LPS (2 μg/ml), **e** qRT-PCR analysis of Il-1β, Il-6, and Tnf transcripts. **f** ELISA detecting cytokines in cultured supernatant. **g** Flow cytometry analyzing surface expression of CD40. **h** C3[−/−] BMDMs were tranfected to overexpress VSIG4, and cells were further treated with LPS (2 μg/ml), the secretion of IL-6 and IL-1β was detected by ELISA. Error bar, s.e.m. *p < 0.05, **p < 0.01, ***p < 0.001 and NS, p > 0.05 (Student's t-test). Data are representative of three independent experiments

maximal OCR (Fig. 4b, c), implying that VSIG4 inhibits mitochondrial oxidation during macrophage activation.

It is believed that glucose oxidation via the mitochondrial electron transport chain is a major source of mtROS upon cells undergoing aerobic metabolism[32]. The fact that VSIG4 inhibited mitochondrial oxidation led us to investigate the status of mtROS, which can induce macrophage M1 activation through activating NF-κB and stabilizing HIF-1α[17]. Compared to the controls, flow cytometry showed that VSIG4+RAW264.7 cells had a significantly less mtROS secretion, especially in response to LPS stimulation (Fig. 4d). Conversely, LPS exposure caused increased mtROS secretion in $Vsig4^{-/-}$ PEMs in vitro (Fig. 4e). Moreover, both ATMs from $Vsig4^{-/-}$ obese mice and PEMs isolated from MHV-3-infected $Vsig4^{-/-}$ mice had higher levels of mtROS than their WT littermates (Fig. 4f). Inhibition of mtROS production by using diphenyliodonium chloride (DPI) can efficiently block LPS-induced IL-6 secretion from both RAW246.7 and VSIG4+RAW264.7 cells (Fig. 4g). This implies that VSIG4 inhibits macrophage M1 activation mainly by reducing pyruvate oxidation and mtROS generation.

The other important consequence of LPS-induced metabolic reprogramming in macrophages is the accumulation of succinate,

a substrate that in turn stimulates HIF-1α-dependent IL-1β expression[13]. Nevertheless, LPS-induced succinate accumulation and HIF-1α upregulation in macrophages appear to be unaffected by the presence of VSIG4 (Supplementary Fig. 7), suggesting VSIG4-mediated macrophage activation is HIF-1α independent.

**VSIG4 enhances PDK2 expression in macrophages.** The observation of VSIG4 inhibiting mitochondrial oxidation and mtROS secretion led us to investigate the underlying molecular mechanisms. Regulation of pyruvate metabolism largely relies on PDH, whose activity is inhibited by PDKs via phosphorylation[33]. In examination of the 4 $Pdk$ isoform expressions in BMDMs and we found that the $Pdk2$ mRNA and protein levels were significantly lower in $Vsig4^{-/-}$ macrophages than that in their WT counterparts (Fig. 5a, b). This low level of PDK2 was responsible for the appreciable decreases in phosphorylation of PDH (p-PDH-E1α$^{S300}$ and p-PDH-E1α$^{S293}$), as detected by western blot and immunofluoresence staining (Fig. 5b, c). The absence of $Vsig4$ in Kupffer cells also resulted in PDK2 reduction and lower p-PDH-E1α$^{S300}$/E1α$^{S293}$ levels in the liver tissues, both in uninfected and at 48 h of MHV-3-infected conditions (Fig. 5d).

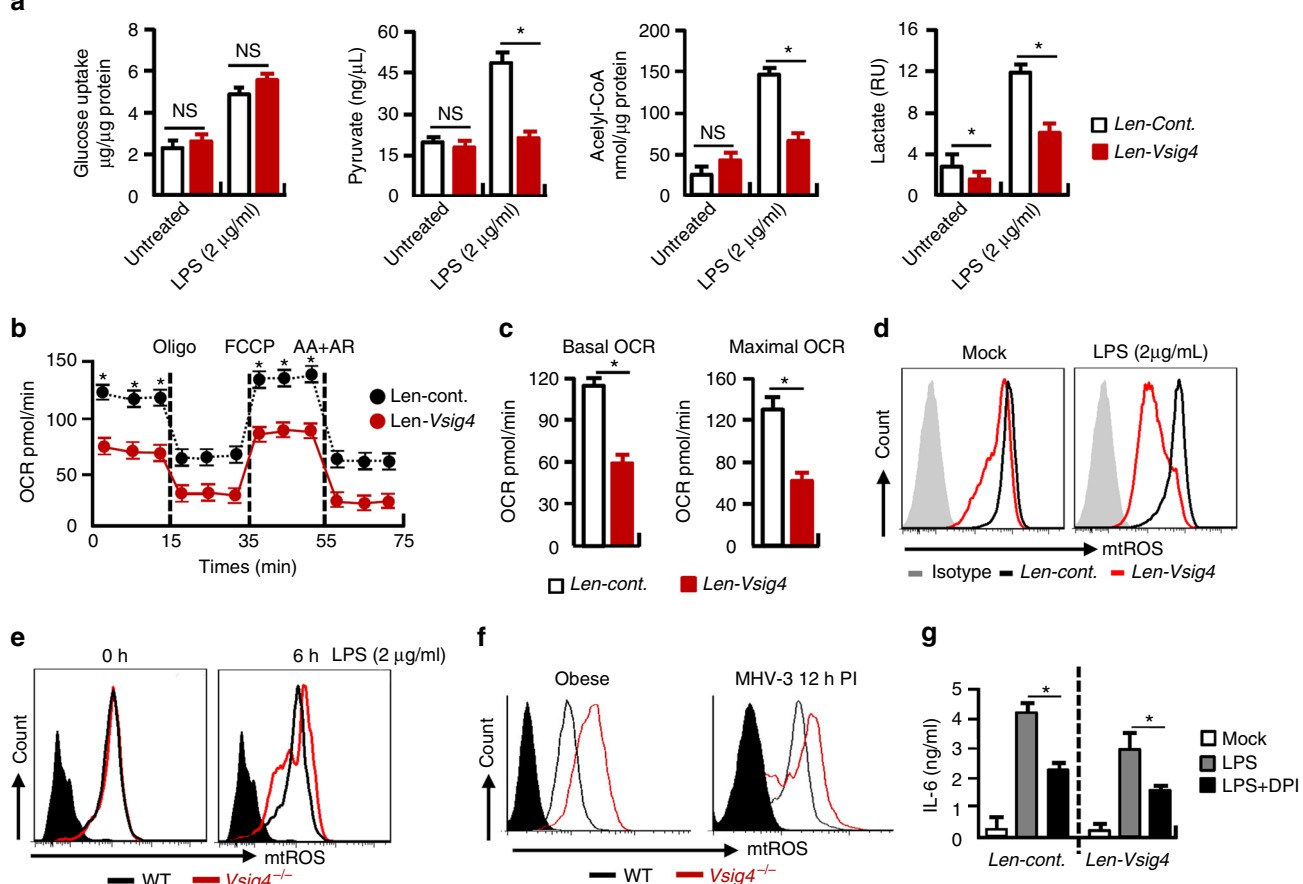

**Fig. 4** VSIG4 reprograms pyruvate metabolism and mtROS generation. VSIG4+RAW264.7 cells and their controls were treated with LPS (2 μg/ml) for 6 h. **a** Colorimetric/Fluorometric assay of glucose uptake, pyruvate, Acelyl-CoA and Lactate levels, $n = 5$ per group. **b** OCR of LPS-treated RAW264.7 cells by Seahorse XFp assay. OCR detected before and after sequential treatment with ATP synthase inhibitor Oligo, mitochondrial uncoupling agent FCCP, ETC inhibitors AA+AR at indicated times, $n = 5$ per group. **c** OCR at basal and maximal levels of the indicated conditions was plotted in bar graphs. **d** mtROS secretion was detected by flow cytometry. **e** $Vsig4^{-/-}$ PEMs and the WT controls were treated with LPS for 0 and 6 h, and mtROS secretion was detected by flow cytometry. **f** mtROS secretion from ATMs of obese mice and PEMs from 12 h of MHV-3-infected animals was compared by flow cytometry. **g** RAW264.7 cells were treated with mtROS inhibitor DPI (10 μM) for 48 h in advance, cells were then added with LPS (2 μg/ml) for an additional 6 h, IL-6 in the supernatant was detected by ELISA. Error bar, s.e.m. *$p < 0.05$ and NS, $p > 0.05$ (Student's $t$-test). Data are representative of three independent experiments

Conversely, transient expression of VSIG4 in RAW264.7 cells appeared to enhance PDK2 expression, and increase PDH-E1$\alpha^{S300}$ phosphorylation, especially after LPS administration (Fig. 5e). Therefore, the PDH activity was decreased (Fig. 5f). These data suggest that VSIG4 promotes PDK2 upregulation in macrophages.

To validate the role of PDK2 in LPS-induced macrophage activation, *Pdk2* expression was silenced in RAW264.7 cells by using the specific shRNA (sh-*Pdk2*). Interestingly, knock-down of *Pdk2* appeared to enhance oxygen consumption, both under normal conditions (Supplementary Fig. 6c, d) and after 2 h of LPS administration (Fig. 5g), implying that PDK2 inhibits mitochondrial oxidation. Moreover, enhancing mitochondrial oxidation caused the elevation of the basal mtROS production, especially after LPS stimulation (Fig. 5h), together with augmenting IL-6 and TNF secretion (Fig. 5i), as well as promoting CD40 expression (Fig. 5j). Similarly, LPS-treated *Pdk2*$^{-/-}$ BMDMs also manifested with enhanced mtROS secretion, promoted the transcription of proinflammatory cytokine genes, as well as enhanced CD40 expression compared to WT counterparts (Supplementary Fig. 8). Conversely, lentiviral overexpression of PDK2 in RAW264.7 cells has opposite effects, not only by quenching of the basal level but also by preventing the LPS-induced upregulation of mtROS (Fig. 5h). This was associated

with a significant reduction in IL-6/TNF secretion and limitation of CD40 upregulation (Fig. 5i, j). Collectively, these results suggest that VSIG4 controls macrophage M1 activation by regulating the PDK2-dependent pyruvate mitochondria metabolic axis.

**VSIG4 promotes DK2 by activating PI3K/Akt–STAT3.** To address the underlying mechanism of VSIG4 promotes PDK2 upregulation in macrophages, we focused on the PI3K–Akt signaling machinery in consideration of the fact that this pathway is essential for cellular metabolism, in addition to other functions such as cell growth, survival etc.[34]. Western blotting data showed that *Vsig4*$^{-/-}$ PEMs and BMDMs had obviously decreased in Akt phosphorylation (Fig. 6a). However, lentivirus-mediated overexpression of VSIG4 in RAW264.7 cells resulted in higher p-Akt$^{ser473}$ expression compared to mock-infected controls (Fig. 6b), suggesting that VSIG4 transfers a feedback signal, licensing macrophages for Akt activation. To identify the functional motifs of VSIG4 that are responsible for augmenting Akt phosphorylation, we created a series of point and truncation mutants of the molecule and tested their activity in affecting Akt activation. The data from probing with p-Akt$^{ser473}$ suggested that the c-terminal residues 267–280 aa of VSIG4 were critical for

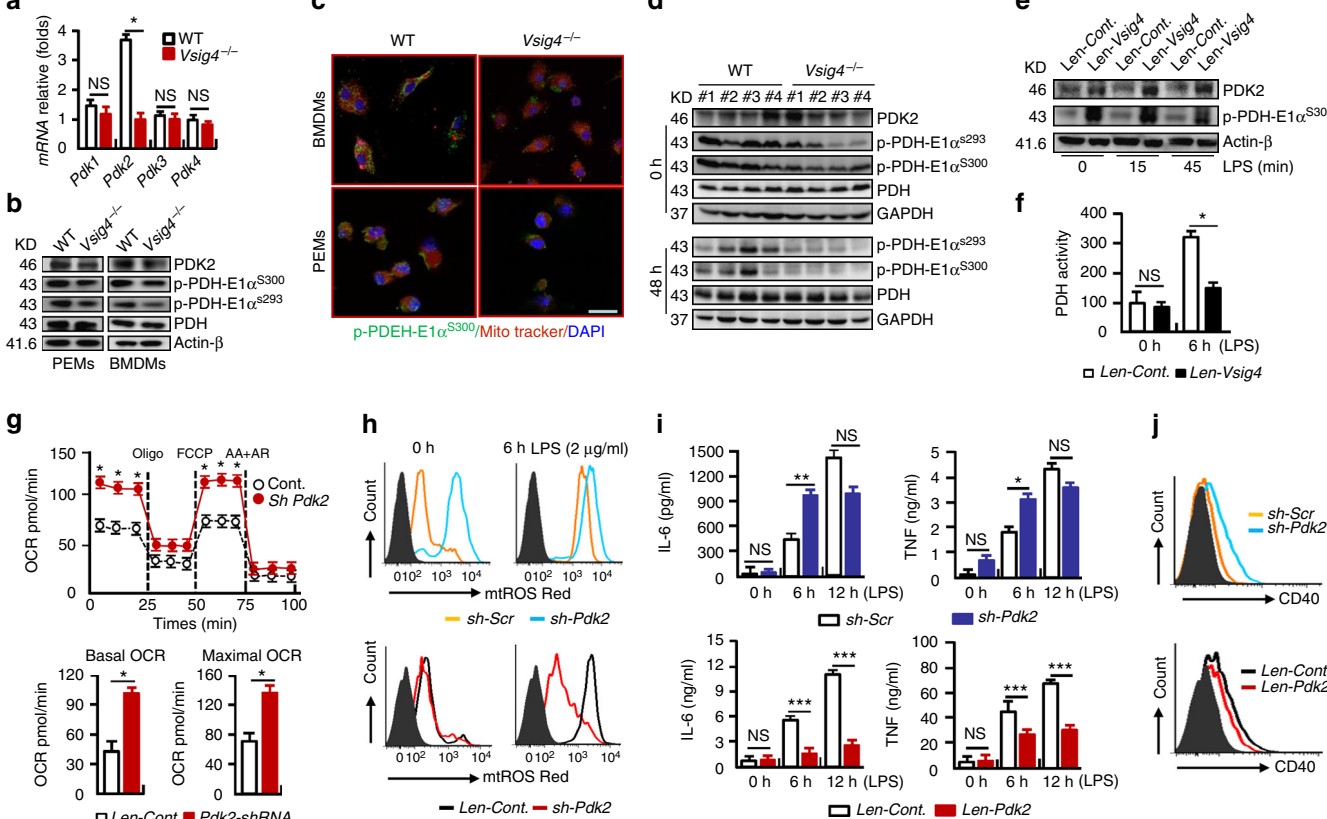

**Fig. 5** VSIG4 triggers PDK2 expression in macrophages. Macrophages from WT and *Vsig4*$^{-/-}$ mice were collected. **a** qRT-PCR detection of 4 *Pdk* isoforms in BMDMs. **b** Western blot analyzing PDK2, p-PDH-E1$\alpha^{S300}$, p-PDH-E1$\alpha^{S293}$, and total PDH. **c** The location of p-PDH-E1$\alpha^{s300}$ in mitochondria was analyzed by immunofluoresence double staining, scale bar = 20µm. **d** Western blot of PDK2, p-PDH-E1$\alpha^{S300}$, p-PDH-E1$\alpha^{s293}$ in liver tissues at 0 h and 48 h PI. RAW264.7 cells were transfected to expression of *Vsig4*, and cells were further treated with LPS (2 µg/ml), **e** Western blot analysis of PDK2 and p-PDH-E1$\alpha^{s300}$. **f** PDH activity analysis, n = 6 per group. The expression of *Pdk2* in RAW264.7 cells was silenced by shRNA or enhancing *Pdk2* expression by lentivirus infection. **g** Seahorse analysis of OCR after 2 h of LPS treatment (up), and basal and maximal OCR of the indicated conditions was plotted in bar graphs (down), n = 5 per group. **h** Flow cytometric assay of mtROS secretion after LPS administration. **i** ELISA of IL-6 and TNF in cultured supernatants, n = 4 per group. **j** Flow cytometric assay of LPS-caused CD40 expression at 6 h. Error bar, s.e.m. *p < 0.05,**p < 0.01, ***p < 0.001 and NS, p > 0.05 (Student's t-test). Data are representative of three independent experiments

mediating Akt phosphorylation (Fig. 6b). Several families of kinases phosphorylate both serine and threonine residues in target substrates, thus result in three dimensional changes of the protein structure and thereby alter its enzymatic activity or affects its ability to interact with other proteins[35]. There are two serine residues (Ser[273] and Ser[276]) and two threonine residues (Thr[270] and Thr[274]) in the c-terminal 267–280 residues of VSIG4, and further assessment revealed that mutation of Thr[270], Ser[273], and Ser[276], but not Thr[274], to Ala respectively, could successfully inhibited LPS-caused Akt[Ser473] phosphorylation in RAW264.7 cells, thus prevented PDK2 upregulation (Fig. 6c), suggesting Thr[270], Ser[273], and Ser[276] residues of VSIG4 play essential role in mediating PDK2 expression. The direct involvement of Akt in modulating PDK2 expression was demonstrated by showing that treating VSIG4+Raw264.7 cells with the Akt inhibitor MK-2206 (Fig. 6d) or the PI3K inhibitor Ly294002 (Fig. 6e), all resulted in typical downregulation of p-Akt and PDK2. These combined data suggest that VSIG4 triggers PDK2 upregulation by activating the PI3K/Akt pathway.

Chromatin immunoprecipitation and massive parallel sequencing (ChIP-Seq) have demonstrated that the Pdk2 promoter region has two binding sites for the signal transduction and activator of transcription-3 (STAT3)[36], which provides a basis for analyzing the signaling pathways responsible for VSIG4-PI3K/Akt-induced PDK2 upregulation. Interestingly, we found that both the PI3K inhibitor Ly294002 (Fig. 6e), and the Akt inhibitor MK-2206 (Fig. 6f), could successfully inhibit LPS-induced STAT3 phosphorylation (p-STAT3) in VSIG4+RAW264.7 cells. Similarly, the STAT3 inhibitor S3I-201 was also able to impair LPS-induced p-STAT3 expression in VSIG4+RAW264.7 cells, thus

resulting in PDK2 downregulation (Fig. 6g). Conversely, silenced Stat3 expression in VSIG4+RAW264.7 cells by specific shRNA also decreased basal PDK2 expression (Fig. 6h). Furthermore, using ChIP-qPCR, we found that LPS induces p-STAT3 recruitment to one of the two putative binding sites at the −1,298 bp but not the −2,934 bp of Pdk2 promoter region, and the present of VSIG4 signaling markedly promotes this recruitment (Fig. 6i). Taken together, these data show that VSIG4 induces PDK2 expression via activating the PI3K/Akt–STAT3 signaling pathway.

To address whether VSIG4 promotes PDK2 expression in macrophage is dependent on C3b, the human monocyte cell line, THP-1 cells, was transfected to overexpress human VSIG4, and these VSIG4+THP-1 cells were further induced to be macrophages by PMA stimulation. Additionally, cells were activated with microbeads-conjugated C3b, and western blot indicates that microbeads-C3b does not affect the basal and LPS-induced PDK2 expression (Fig. 6j). Furthermore, overexpress VSIG4 in C3−/− BMDMs also increased basal and LPS-induced PDK2 expression (Fig. 6j). These combined data suggest that VSIG4-mediated PDK2 upregulation in macrophages is C3b independent.

**Promoter methylation inhibits Vsig4 gene transcription.** Our data indicate that VSIG4 suppresses macrophage-dependent inflammation by augmenting PDK2 expression, which highlights a plausible therapeutic intervention for inflammatory disorders through enhancing VSIG4 signaling. However, in agreement with the previous report[20], we found that the PEMs and liver tissues isolated from MHV-3-infected mice manifested

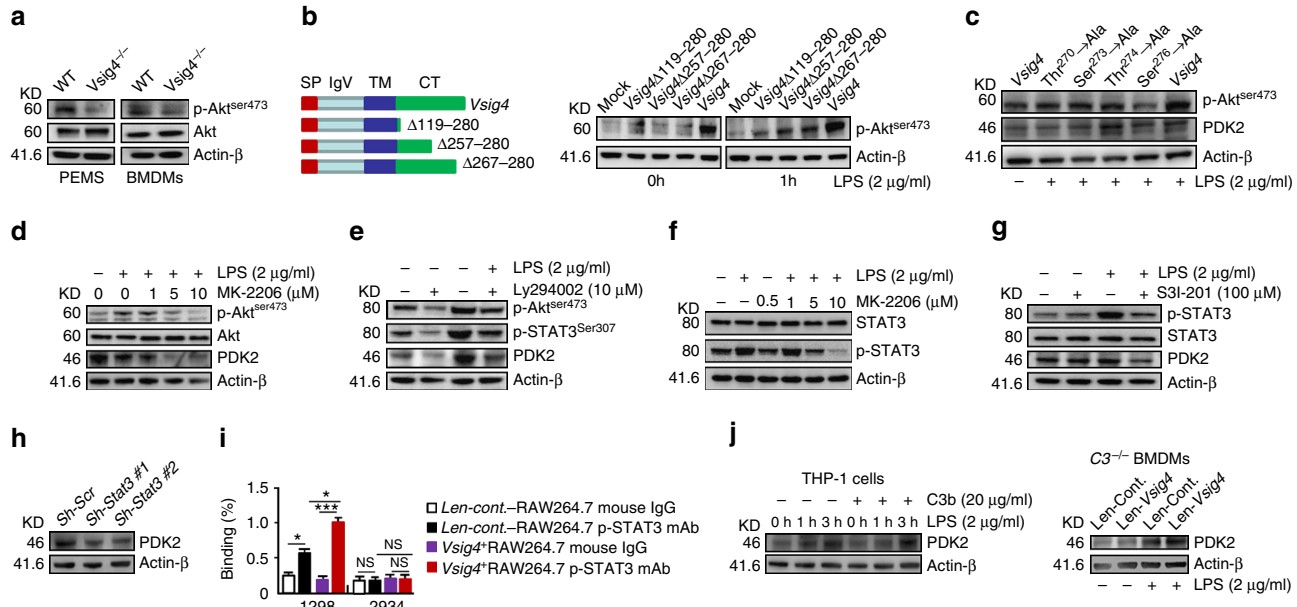

**Fig. 6** VSIG4 promotes PDK2 expression through activating PI3K/Akt–STAT3. **a** Western blot of Akt and p-Akt[ser473] expression. **b** RAW264.7 cells were infected with different Vsig4 deletion constructs, cells were further treated with LPS (2 μg/ml), the expression of p-Akt[ser473] was analyzed by western blot. **c** Western blot of p-Akt[ser473] and PDK2 in VSIG4+RAW264.7 cells with mutation of Ser[273], Ser[276], Thr[270], and Thr[274] to Ala. VSIG4+RAW264.7 cells were treated with **d** the Akt inhibitor MK-2206, **e** the PI3K inhibitor Ly294002, and the expression of Akt, p-Akt[ser473], PDK2, and p-STAT3 was detected by western blot. **f** VSIG4+RAW264.7 cells were treated with the MK-2206, and the expression of STAT3/p-STAT3 was analyzed by western blot. **g** Western blot of p-STAT3/STAT3 and PDK2 in LPS-activated VSIG4+RAW264.7 cells followed with STAT3 inhibitor, S3I-201 (100 μM) treatment for 24 h. **h** Western blot of PDK2 in Stat3 silenced VSIG4+RAW264.7 cells. **i** The enrichment of p-STAT3 in Pdk2 gene promoter region was detected by ChIP-qPCR assay. **j** Human VSIG4+THP-1 cells were treated with microbeads-C3b (20 μg/ml) in the presence of LPS (2 μg/ml), and the expression of PDK2 was detected by western blot. Moreover, C3−/− BMDMs were transfected to overexpress VSIG4, and cells were further treated with LPS (2 μg/ml) for an additional 3 h, and the expression of PDK2 was detected by western blot. Error bar, s.e.m. *p < 0.05, ***p < 0.0001 and NS, p > 0.05 (Student's t-test). Data are representative of three independent experiments

with lower VSIG4 expression as compared to the uninfected controls (Fig. 7a, b). Moreover, administration of proinflammatory mediators (including LPS, TNF, MALP2, IFN-γ, poly (I:C) or CpG) apparently can induce a transient sharp decline of *Vsig4* gene transcription and protein in ex vivo PEMs (Fig. 7c, d). Thus, rapid VSIG4 downregulation appears to be a common response of macrophages upon inflammatory stimulations.

The epigenetic mechanisms, especially DNA methylation of CpG sites within promoter regions, have recently been described to mediate gene silencing[37]. There are three types of mammalian DNA methyltransferases (Dnmt), distinct from Dnmt1 that is responsible for copying DNA methylation patterns during replication, Dnmt3a and Dnmt3b are important in de novo DNA methylation[37]. We thereafter hypothesized that inflammation leads to silencing *Vsig4* gene transcription through triggering the transcriptional activation of Dnmts. Western blot showed that the expression of Dnmt3a was upregulated in BMDMs in response to proinflammatory stimuli (Fig. 7e), nevertheless, the expression of Dnmt1 and Dnmt3b appeared to be not affected under such conditions (Supplementary Fig. 9). However, 5-aza-2′-deoxycytidine (AZAdC), a general DNA methyltransferases inhibitor, was able to effectively downregulate Dnmt3a expression, and in agreement, macrophages treated with AZAdC appeared to be resistant to the proinflammatory factors-caused VSIG4 downregulation (Fig. 7e).

To validate the importance of methylation in controlling *Vsig4* gene transcription, a 840 bp fragment of *Vsig4* gene promoter (−840/ + 1) was constructed into a luciferase reporter pGL3-basic vector. The pGL3-Basic and *Vsig4* promoter constructs (−840/ + 1) were further fully methylated by CpG Methyltransferase (M. SssI). We transfected RAW264.7 cells with these plasmids and compared their reporter luciferase activities. Interestingly, the luciferase activity of the M.SssI-methylated construct exhibited 53% reduction in promoter activity, indicating DNA methylation

negatively regulates *Vsig4* gene transcription (Fig. 7f). In accordance, proinflammatory stimuli seemed to have no additional inhibition on the promoter activity in RAW264.7 cells transfected with M.SssI-methylated constructs (Fig. 7f). These data demonstrate that Dnmt3a controls *Vsig4* gene repression through fast methylation of *Vsig4* gene promoter.

We also analyzed the genomic DNA sequences of isolated BMDMs that were treated with various proinflammatory mediators for 12 h. Proinflammatory stimuli all appeared to induce very high incidence (98–100%) of methylation at a CpG site (-374 bp) in the promoter region of *Vsig4* gene, which is significantly elevated from a 75% basal methylation at this site in the untreated controls as detected by using the Sequenom MassARRAY platform (Supplementary Fig. 10a, b, Supplementary Table 1). Furthermore, ChIP-qPCR assays reveal a significant enrichment of Dnmt3a in −374 bp of *Vsig4* promoter region after the BMDMs were treated with proinflammatory factors (Supplementary Fig. 10c). These combined data suggest that CpG at -374 bp site, probably with other CpG islands in the *Vsig4* promoter region, play an essential role for negative feedback control of macrophage activation during inflammatory response.

**Forced overexpression of *Vsig4* improves MHV-3-caused FH.** We therefore tried to transiently force the expression of *Vsig4* in MHV-3-susceptible C57BL/6 WT mice using lentiviral vectors in vivo. These mice expressed significantly higher levels of VSIG4 in the livers on day 6 of transduction compared to control infected animals (Fig. 8a). Interestingly, *Vsig4*-transgenetic mice had significant enhancing PDK2 expression while lessening PDH phosphorylation (p-PDH-E1α$^{S300}$) in liver tissues at 72 h of MHV-3 infection (Fig. 8b), leading to lower levels of FGL2, TNF, IL-1β, and IL-6 deposition in liver tissues (Fig. 8b), together with reducing liver damage (Fig. 8c), along with a considerably

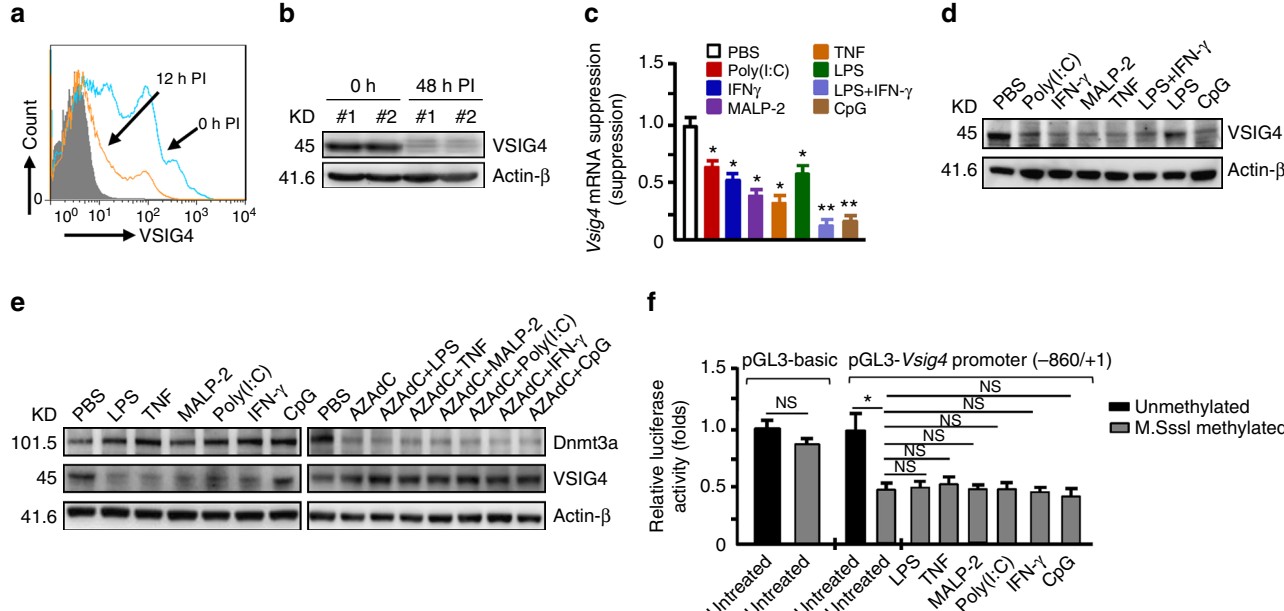

**Fig. 7** *Vsig4* gene transcription is repressed Dnmt3a-mediated DNA methylation. The C57BL/6 WT mice were infected with MHV-3 (100 PFU/mouse), **a** the expression of VSIG4 on PEMs at 0 h and 12 h PI was detected by flow cytometry. **b** VSIG4 protein level in liver tissues was analyzed by western blot. The BMDMs were treated with proinflammatory factors for 12 h. **c** *Vsig4* gene transcription was detected by qRT-PCR. **d** VSIG4 protein expression was evaluated by western blot. **e** The BMDMs were treated with Dnmts inhibitor-AZAdC (10 μM) for 72 h in advance, cells were then further added with proinflammatory mediators for 12 h, the expression of Dnmt3a and VSIG4 was assessed by western blotting. **f** Luciferase activity of the lysates from RAW264.7 cells transfected with unmethylated or M.SssI methylated pGL3-basic vector and the -840/+1 *Vsig4* promoter constructs. Error bar, s.e.m. *$p <$ 0.05, **$p <$ 0.01 and NS, $p >$ 0.05 (Student's *t*-test). Data are representative of three independent experiments

improved survival rate (Fig. 8d). These combined data suggest that increasing the expression of VSIG4 might have therapeutic potentials for fulminant hepatitis and other macrophage-associated inflammatory disorders (Supplementary Fig. 11).

## Discussion

Macrophage activation relies on metabolic adaptation in response to the surrounding micro-environmental stimuli. Macrophage plasticity determines its biological functions in immunity, inflammation, and tissue homeostasis. Defining the mechanisms regulating macrophage metabolic patterns is critical for understanding the pathology of inflammatory disorders and developing therapeutic interventions[2]. We here demonstrate that VSIG4, a B7 family-related protein that is expressed specifically in resting macrophages, is able to inhibit macrophage activation by reprogramming mitochondrial pyruvate oxidation. *Vsig4* deficiency apparently sways macrophage towards activation upon LPS exposure in vitro. Conversely, overexpression of *Vsig4* suppresses M1 gene expression and reduces LPS-induced pyruvate oxidation and mtROS secretion by RAW264.7 cells. Interestingly, *Vsig4* deficiency affects the outcomes of inflammatory disorders in animal models. For instance, $Vsig4^{-/-}$ mice are more susceptible to develop HFD-induced obesity and insulin resistance. Furthermore, $Vsig4^{-/-}$ mice exhibit markedly higher mortality over MHV-3 viral infection, clearly due to the exacerbated macrophage-dependent inflammation in vivo. Finally and the most notably, the anti-inflammatory function of VSIG4 has been validated by forced overexpression of VSIG4 in vivo, for which we show that excessive epigenetic VSIG4 dampens liver tissue inflammation and protects the susceptible mice from MHV-3 virus-induced FH. These studies demonstrate that VSIG4 inhibits macrophage M1-associated inflammatory pathogenesis.

It has long been known that ROS plays essential roles in immune responses of macrophages and neutrophils to pathogens. The bacteria killing capability of activated macrophage and neutrophils is due to reduction in cellular nicotinamide adenine dinucleotide phosphate oxidases activity that results in production of superoxide during the respiratory burst. On the other hand, over production of ROS actively participates in the pathogenesis of inflammatory diseases including rheumatoid arthritis, multiple sclerosis, and thyroiditis through activating the inflammatory signaling pathways including mitogen-activated protein kinases (MAPK), NF-κB, and guanylate cyclase[38]. ROS comes from various sources, such as peroxisomes, ubiquinone, activities of cytosolic enzymes and uncoupled nitric oxide synthases. However, recent data identify that mitochondria are a major source of physiological intracellular ROS that drives inflammation[39]. mtROS can be sensed by the NLRP3 inflammasome, resulting in caspase-1 activation and IL-1β maturation[15]. Therefore, high levels of mtROS in vivo would impair glucose metabolism and insulin sensitivity[16], and it probably explains why exaggerated macrophage activation promotes pathogenesis of MHV-3-mediated hepatitis[29]. Alternatively, mtROS also activates NF-κB and stabilizes HIF-1α, by thus it increases the activities of these two transcriptional factors leading to upregulation of macrophage M1 genes[18]. Here, we show that VSIG4 can inhibit pyruvate/acetyl-CoA conversation in RAW264.7 cells, leading to limitation of oxygen consumption (Fig. 4a–d). Interestingly, decreasing mitochondrial oxidation also leads to the inhibition of LPS-induced mtROS secretion, along with restriction of *Il1b*, *Il6*, and *Tnf* gene transcription and CD40 upregulation (Fig. 3). Moreover, PEMs isolated from MHV-3-infected $Vsig4^{-/-}$ mice or ATMs collected from HFD-fed $Vsig4^{-/-}$ obese animals all exhibited with more mtROS secretion in vivo as compared to WT littermates (Fig. 4f). Similar to previous studies[40], inhibition of mtROS activity by DPI successfully prevented LPS-mediated IL-6 production (Fig. 4g). These data suggest that VSIG4 inhibits macrophage activation via posing restriction on mitochondrial oxidation and mtROS secretion.

mtROS derives from pyruvate metabolism during OXPHOS in the mitochondria, in which the conversion of cytosolic pyruvate

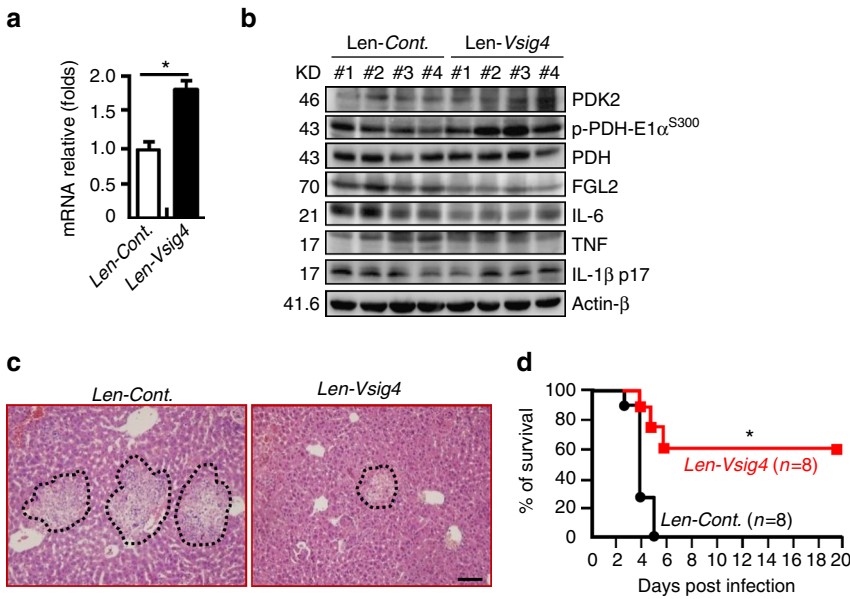

**Fig. 8** Forced overexpression of *Vsig4* improves MHV-3-induced hepatitis. C57BL/6 WT mice were infected with lentivirus (10[7] PFU/mouse) to induce the expression of Vsig4 in vivo, these mice were further infected with MHV-3 at day 6. **a** Liver *Vsig4* gene transcription was analyzed by qRT-PCR at day 6, n = 5 per group. **b** Western blotting for PDK2, p-PDH-E1α[s300], FGL2, and proinflammatory cytokines TNF, IL-6 and IL-1β in liver tissues at 72 h of MHV-3 infection, n = 4 per group. **c** The architecture of the liver tissues at 72 h of infection was compared by H&E staining, scale bar = 20 μm, n = 5 per group. **d** The survival was monitored for a total of 20 days. Error bar, s.e.m. **a** *p < 0.05 was analyzed by Student's t-test, and **d** was analyzed by log-rank test. Data are representative of three independent experiments

into mitochondrial acetyl-CoA is partially regulated by the activity of PDH[41]. PDH activity is enhanced via its dephosphorylation by phosphopyruvate dehydrogenase phosphatase (PDP), and inducing PDP1 expression has been shown to promote mitochondrial OXPHOS, mtROS production and M1 gene expression[19]. Conversely, PDK-mediated phosphorylation of the PDH-E1α subunit at 3 serine sites, including $Ser^{232}$, $Ser^{293}$, and $Ser^{300}$, appeared to cause PDH suppression[42]. Recent studies have shown that PDK/PDH axis essentially controls cytokine secretion in macrophages. For example, overexpression of PDK2 inhibits radiation-induced cytokine expression[43], whereas macrophages from *Pdk2/4*-deficient animals appear to produce less IL-1β and TNF in association with a low level of lactic acid[44]. We here identify that the VSIG4-dependent inhibition of macrophage M1 activation relies on PDK2. LPS stimulation resulted in augmentation of mtROS secretion and inflammatory gene expression in *Pdk2* silenced RAW264.7 cells and *Pdk2* deficient BMDMs (Fig. 5h–j, Supplementary Fig. 8). In agreement, overexpression of *Pdk2* results in opposite effects (Fig. 5h–j). Along with previous work suggesting that PDK1 is involved in promoting aerobic glycolysis in macrophages[45], these combined data highlight a signaling pathway that governs the fate of macrophages differentiation and function and *Pdk* status is involved in mediating metabolic programs for balancing between glycolysis and glucose oxidation.

Substantial evidence demonstrates that obesity is a chronic low-grade inflammatory disease[46]. In obesity, adipocytes can release proinflammatory mediator like CC chemokine ligand (CCL)-2, and monocyte chemoattractant molecule (MCP)-1, which induce the recruitment of ATMs[47]. ATM secretes proinflammatory cytokines and forms the inflammatory circuit which blocks the insulin action in adipocytes and leads to insulin resistance[5]. Here we demonstrate that ATMs isolated from HFD-fed *Vsig4*[−/−] obese mice have higher levels of proinflammatory factors such as TNF, IFN-γ, and IL-1β (Fig. 1i–k). It was elucidated that proinflammatory cytokines promote obesity-associated insulin resistance through activating inflammatory signaling pathways, including the stress-responsive c-Jun NH2-terminal kinase (JNK1/2) and AMP-activated protein kinase α2 (AMPKα2), inhibitor of κB kinase, and extracellular signal-regulated kinase 1/2 (ERK1/2) as well as MAPKp38, collectively causing inhibition of serine/threonine phosphorylation of the docking protein IRS-1[48–50]. In agreement with previous studies, we found that *Vsig4*[−/−] obese mice exhibit exacerbated insulin resistance along with diminished p-IRS-1 and p-Akt[ser473] in the WAT and the liver tissues (Fig. 1h), indicating that *Vsig4* deficiency eliminates the negative control signals, thus allowing ATMs to produce proinflammatory cytokines and switching on HFD-associated insulin tolerance in vivo.

Microorganism infection and tissue injuries trigger the recruitment of inflammatory macrophages from the circulation into the affected tissues[51]. The MHV-3 virus provokes a mouse strain-associated severe liver disease that has been used as a model for investigating human viral fulminant hepatitis. This exacerbation of macrophage-dependent cytokine storm directly causes hepatic necrosis and induces lethality in susceptible mouse strains[7]. In support, normal induction of macrophage apoptosis can appropriately control local liver tissue fibrinogen deposition and local tissue injuries in this fulminant hepatitis model[52]. As expected, we have found that MHV-3 infection super-induces production of a panel of inflammatory cytokines from *Vsig4*[−/−] macrophages, and the virus-infected mice exhibit with more severe liver necrosis and high mortality (Fig. 2). In contrast, forced expression of *Vsig4* in susceptible mice provides significant protection against MHV-3-induced fulminant hepatitis (Fig. 8). These combined results demonstrate that VSIG4 functionally contributes to limiting macrophage M1-associated inflammation, and its deficiency can cause over inflammatory disorders.

Identification of mediators that regulate *Vsig4* expression will bring insights into understanding the inflammatory response in the host. The epigenetic mechanisms, such as DNA methylation and histone modification, have recently been shown to mediate gene silencing[53]. DNA methylation is a covalent modifications, in which DNA methyltransferase catalyze cytosine methylation using S-adenosyl methionine as a methyl donor[54]. It has been reported that the expression of tumor suppressor genes in several human cancers were inhibited upon DNA methylation at their CpG sites within the promoters[37, 55]. We here found that *Vsig4* mRNA and protein were decreased dramatically in PEMs that were treated with inflammatory factors in vitro (Fig. 7c–e). Conversely, blocking Dnmt3a activity by the inhibitor, 5-aza-2′-deoxycytidine, could effectively prevent proinflammatory factors induced *Vsig4* downregulation (Fig. 7e). Further studies demonstrated that the luciferase activity of the M.SssI-methylated *Vsig4* promoter (-840/+1) was reduced by 53% comparing to the unmethylated controls, and proinflammatory stimuli were unable to further inhibit the promoter activity in RAW264.7 cells transfected with the M.SssI-methylated *Vsig4* promoter constructs (Fig. 7f). We also found that some CpG sites in *Vsig4* promoter region (like CpG at -374 bp site) were methylated by proinflammatory stimuli as detected by using the Sequenom MassARRAY platform (Supplementary Fig. 10, Supplementary Table 1). Together, these data suggest that proinflammatory factors repress *Vsig4* gene transcription through inducing Dnmt3a activity, causing fast methylation of *Vsig4* promoter region, for which CpG at -374 bp site, probably with other CpG islands within the *Vsig4* promoter region, plays a critical role in regulating *Vsig4* transcription.

Recent studies have illustrated that VSIG4 suppresses T-cell cytokine production and causes cell cycle arrest. Injection of soluble VSIG4-Ig protein causes a reduction in IFN-γ production by antigen-specific CD8[+] T cells and down-modulates Th-dependent IgG responses in vivo[20]. Additionally, administration of VSIG4-Ig protein prolonged mouse survival in a ConA-induced hepatitis model and protected against the pathogenesis of many inflammatory diseases[25, 56]. One of the explanations for this effect is that VSIG4 binds to an as-yet-unidentified inhibitory receptor/ligand on T cells. Alternatively, the surface VSIG4 on macrophages and CR1/CR3 on T cells bind to the same multimeric C3b or iC3b molecules, thus triggering signals to suppress T-cell activation[57]. Here, we showed that the secretion of cytokines in T cells that isolated from liver tissues of 72 h post MHV-3-infected *Vsig4*[−/−] animals were similar to their WT counterparts (Supplementary Fig. 3), implicating VSIG4 does not affect T-cell activation in our models. Conversely, we demonstrate that VSIG4 delivers negative feedback signals to macrophages, resulting in PI3K–Akt–STAT3 dependent PDK2 upregulation, and finally impedes mtROS-dependent M1 macrophage activation, suggesting VSIG4 negatively controls macrophage activation. However, microbeads-C3b did not affect the expression of PDK2 from LPS-activated VSIG4[+]THP-1 cells. Additionally, the expression of PDK2 and the secretion of cytokines like IL-6 as well as IL1-β were still downregulated in LPS-stimulated *C3*[−/−] BMDMs (Figs. 3h, 6j), these combined data suggest that inhibition of macrophage activation by VSIG4 is C3b independent. Further studies are needed to identify the potential interacting partners for VSIG4.

In summary, we have demonstrated that VSIG4 down-modulates macrophage activation and M1 polarization in response to inflammatory stimuli in vitro and in vivo. Mechanistically, VSIG4 is capable of sending feedback signals in macrophages to activate the PI3K–Akt–STAT3 signaling axis, leading

to PDK2 upregulation and activation, thus inhibiting mitochondrial pyruvate metabolism, suppressing mtROS secretion and M1-like gene expression through inducing PDH phosphorylation. Therefore, we speculate that enhancing VSIG4 signaling may result in beneficial effects on treating inflammatory disorders.

## Methods

**Mice.** The complement C3 deficient ($C3^{-/-}$) mice (#003641) and the C57BL/6 mice were purchased from Jackson Laboratory. The $Vsig4^{-/-}$ mice were kindly provided by Dr. M. van Lookeren Campagne (Department of Immunology, Genentech, CA, USA). The $Pdk2^{-/-}$ mice were provided by Dr. C.R. Harris (Rutgers Cancer Institute of New Jersey, USA). All mice were backcrossed ten times onto the B6 background to avoid unpredictable confounders. Specific pathogen-free male and age-matched mice (8–12 weeks old) were used for all experiments. Mice were maintained in micro-isolator cages, fed with standard laboratory chow diet and water, and housed in the animal colony at the animal center of the Third Military Medical University (TMMU). All animals received humane care according to the criteria outlined in the "Guide for the Care and Use of Laboratory Animals" prepared by the National Academy of Sciences and published by the National Institutes of Health (NIH publication 86–23 revised 1985). All of the in vivo experiments comply with the animal study protocol approved by the ethics committee of TMMU.

**Cells.** The mouse macrophage cell line RAW264.7, human monocyte cell line THP-1 and 293T cells were provided by the Cell Institute of the Chinese Academy of Sciences (Shanghai, China). Mouse 17 clone 1 (17CL1) cells were purchased from ATCC. The cells were cultured in 6-well plates and propagated in DMEM supplemented with 10% FBS, 100 U/ml penicillin, and 100 μg/ml streptomycin. Peritoneal exudative macrophages (PEMs) were harvested and BMDMs induced by M-CSF (#400-28, Peprotech, Rocky Hill, NJ, USA) as described previously[52].

**Virus and infection.** MHV-3 viruses were expanded in 17CL1 cells to a concentration of $1 \times 10^7$ plaque forming units (PFU)/ml. Mice were received MHV-3 (100 PFU/mouse) via intraperitoneally (i.p.) injection. In some experiments, C57BL/6 WT mice were infected with lentivirus ($1 \times 10^7$ PFU/mouse) via intravenous injection to transiently force the expression of Vsig4 in vivo, and these mice were further infected with MHV-3 after 6 days. The virus-infected mice were euthanized on the indicated days. Liver damage was compared by H&E staining and cell apoptosis was measured using the Terminal Transferase dUTP Nick End Labeling (TUNEL) staining method according to the manufacturer's instructions (#12156792910, Roche, Mannheim, Germany). The virus titers in liver tissues were determined by plaque assay method. Briefly, the supernatant of liver tissue homogenate was ten times step diluted. Mouse 17CL1 cells were seeded in 12-well plates, when reaches 80% fusion, cells were added with the diluted liver extraction and incubated for 30 min under 37 °C, 5% CO$_2$ condition. Cells were then added with 1 ml 2% MethyCellulose DMEM medium (100 U/ml penicillin and 100 μg/ml streptomycin) per well, and then further incubated for 4 days. Cells added with the supernatant from uninfected liver tissue homogenate were used as negative controls, whereas cells treated with the purified MHV-3 virus were used as positive controls. Finally, cells were fixed with 4% paraformaldehyde and the viral titers were determined by crystal violet staining assay.

**Diet intervention.** C57BL/6 WT mice and congenic $Vsig4^{-/-}$ littermates were received NCD or HFD (#MD12031, Medicience Ltd., Nanjing, China) starting at the age of 8 weeks. The body mass was evaluated every week for a total of 10 weeks. The distribution of fat tissues in obese mice was made using an in vivo microcomputed tomography scanner (μCT, Quantum FX, Perkin Elmer, Hopkinton, MA, USA). At the end of the feeding experiment, the mice were sacrificed, and blood was collected in EDTA-coated tubes and centrifuged to collect plasma. The liver and epididymal WAT were dissected, weighed, and immediately frozen in liquid nitrogen. The morphometry of individual fat cells was assessed using digital image analysis as described previously[26]. Briefly, microscopic images were digitized in 24-bit RGB (specimen level pixel size $1.28 \times 1.28$ μm$^2$). Recognition of fat cells was initially performed by applying a region growing algorithm on manually indicated seed points, and the minimum Feret diameter was calculated.

**GTT and ITT experiments.** For GTT, the animals were i.p. injected with 2 g/kg glucose (#G6125, Sigma-Aldrich, St. Louis, MO, USA) after 12 h of fasting, and blood was drawn to measure blood glucose 0, 15, 30, 60, 90 and 120 min after injection. For ITT, 0.5 U/kg of insulin (Novolin R, Novo Nordisk) was i.p. injected after 6 h of fasting, and blood was drawn at 0, 15, 30, 45, and 60 min thereafter.

**Seahorse XFp metabolic flux analysis.** The OCR was measured using an XFp extracellular analyzer (Agilent Technologies, Santa Clara, CA, USA). Macrophages were seeded at $2.0 \times 10^4$ cells/well density in 8-well plates for 5 h to allow adherence to the plate. After 2 h of LPS (800 ng/ml) administration, the cells were changed to unbuffered assay media (base medium supplemented with 10 mM glucose, 1 mM

pyruvate, 2 mM glutamine, pH 7.4) and incubated in a non-CO$_2$ incubator for 1 h. Four baseline measurements were taken before sequential injection of mitochondrial inhibitors oligomycin, FCCP, and antimycin (AA) plus rotenone (AR) provided by the manufacturer (#101706-100, Agilent Technologies). OCR was automatically calculated using the Seahorse XFp software. Every point represents an average of three different wells.

**Immunohistochemistry and immunofluorescence double staining.** Paraffin-embedded tissue blocks were cut into 2.5 μm slices and were mounted on polylysine-charged glass slides. Endogenous peroxidase activity was blocked by exposure to 3.0% H$_2$O$_2$ for 30 min. Antigen retrieval was performed in a citrate buffer (pH 6.0) at 120 °C for 10 min. Sections were then incubated at 4 °C overnight with anti-mouse FGL2 (#sc-100276, Santa Cruz, 1:100, mouse), anti-Fibrinogen (#ab118533, Abcam, Cambridge, England, 1:1000, Rabbit), anti-pro-IL-1β (#12242, Cell Signaling Technology (CST), 1:100, mouse), anti-TNF (#3707, CST, 1:100, rabbit), anti-IL-6 (#sc-130326, Santa Cruz, 1:200, mouse), and anti-IFN-γ (#sc-52557, Santa Cruz, 1:200, rat). After washing, the sections were incubated with the corresponding secondary antibodies for 2 h at room temperature. The Vecta-stain ABC kit (Vector Laboratories, San Diego, CA, USA) was used to perform the avidin–biotin complex method according to the manufacturer's instructions. Sections incubated with isotype and concentration matched immunoglobulins without primary antibodies were used as isotype controls. Peroxidase activity was visualized with the DAB Elite kit (K3465, DAKO), and brown coloration of tissues represented positive staining.

To detect p-PDH-E1α$^{S300}$ and mitochondria co-localization, both BMDMs and PEMs were fixed with 4% paraformaldehyde, permeabilized with 0.1% Saponin in PBS for 5 min, and blocked with PBS containing 2% BSA for 1 h at 4 °C. The cells were then stained with rabbit anti-p-PDH-E1α$^{S300}$ antibodies (#AP1046, 1:100, Merk, Temecula, CA, USA) overnight at 4 °C, and then stained with Alexa Fluor488-conjugated donkey anti-rabbit IgG (H+L) highly cross-adsorbed secondary antibody (#A-21206, Thermo Scientific, Billerica, MA, USA) for 1 h. Finally, the sections were incubated with 1 μg/ml DAPI and MitoTracker Red (#M-7512, Thermo Scientific) at 500 nM for 30 min at 37 °C. The results were analyzed using fluorescence microscopy (Zeiss Axioplan 2).

**ELISA and western blotting.** The concentration of cytokines, free fatty acid, and triglycerides in the serum or the culture supernatants, the levels of succinate, pyruvate, acetyl-CoA, triglyceride and PDH enzyme activity in macrophages were measured by the ELISA according to the manufacturer' introductions. ELISA Kits, including TNF (#EK0527), IL-6 (#EK0411), IL-1β (#EK0394), and IL-12 p40 (#EK0932) were from Boster Ltd. (Wuhan, China). FGL2 ELISA Kit was from Uscn Life Science (#SEA512Mu, Wuhan, China). Insulin ELISA Kit was from Millipore (#2617704, Billerica, MA, USA). Pico ProbeAcetyl-CoAFluorometric Assay kit was from BioVision (#K317-100, Milpitas, CA, USA). The Free Fatty Acid Quantitation Kit (#MAK044-1KT), PDH Activity Assay Kit (#MAK183-1KT), Pyruvate Assay Kit (#MAK071-1KT), Lactate Assay Kit (#MAK064-1KT), Succinate Colorimetric Assay Kit (#MAK184-1KT), and other chemicals were all from Sigma-Aldrich (St. Louis, MO, USA).

The expression of GAPDH (#2118, CST, Danvers, MA, USA,1:1000, rabbit), PDK2 (#sc-14486, Santa Cruz, 1:500; goat), p-IRS-1(p-Ser$^{307}$, #2381, CST, 1:1000, rabbit), Akt (#2920, CST, 1:1000, mouse), p-Akt (p-Thy$^{308}$, #4056; p-Ser$^{473}$, #4051, CST, 1:1000, rabbit), PDH (#3025, CST, 1:1000, rabbit), p-PDH-E1α$^{293}$ (#AP1062, Merk, 1:2000, rabbit), p-PDH-E1α$^{S300}$ (#AP1046, Merk, 1:2000, rabbit), STAT3 (#9139, CST, 1:1000, mouse), p-STAT3 (p-Tyr$^{705}$, #9145, CST, 1:2000, rabbit), FGL2 (#sc-100276, Santa Cruz, 1:500, mouse), TNF (#3707, CST, 1:1000, rabbit), pro-IL-1β (#12242, CST, 1:1000, mouse), IFN-γ (#sc-52557, Santa Cruz, 1:200, rat), and IL-6 (#sc-130326, Santa Cruz, 1:200, mouse) in macrophages or liver tissues as well as VAT was measured by western blot. Uncropped western blot images are shown in Supplementary Fig. 12.

**Flow cytometry.** To measure the mROS superoxide, macrophages were incubated with MitoSOX red (5 μM, Life technologies, Eugene, Oregon, USA) at 1.0 μM for 1 h in phenol red-free DMEM (Invitrogen). The death cells were excluded firstly by staining with LIVE/DEATHFixable Near-IR Ded Cell Stain Kit (Life Technologies). To measure the expression of activation markers on cell surface, suspended cells were incubated for 1 h at room temperature in dark using fluorescent antibodies (anti-B7-H1, anti-B7-DC, anti-B7-H3, and anti-CD40). To detect intracellular proinflammatory cytokines (pro-IL-1β, TNF, and IL-6) expression, macrophages or T cells were isolated and treated with brefeldin A for 4 h. mAbs were then added, and further incubated for an additional 1 h. all of these fluorescent antibodies were purchased from eBioscience (San Diego, CA, USA). A total of 10,000 live cells were analyzed by FACsAria cytometer (BD, Franklin Lakes, NJ, USA). All the flow cytometry data were analyzed using CellQuest Pro software.

**Quantitative RT-PCR.** Total RNA was extracted from cultured cells or the indicated tissues with TRIzol reagent according to the manufacturer's instructions (Invitrogen). First-strand cDNA was synthesized with the PrimeScript RT-PCR Kit (Takara, Dalian, China). The expression of mRNA encoding for the indicated genes was quantified by quantitative (q)RT-PCR with the SYBR Premix ExTaq kit

(Takara) and was normalized to the expression of β-actin. qRT-PCR was performed with specific primers (Supplementary Table 2). The results were compared by the $2^{-\Delta\Delta Ct}$ method.

**Lentiviral constructs and transduction.** The mouse Vsig4 (NM_177789) cDNA ORF clone (#MR203780) and Pdk2 (NM_133667) cDNA ORF clone (#MG206400) were purchased from OriGene Technologies, Inc. (Rockville, MD, USA). The whole gene expression cDNA for Vsig4, Pdk2, the truncation mutants of Vsig4, and Vsig4 site-directed mutagenesis were further amplified with specific primers (Supplementary Table 3). cDNA was cloned into the pCDH-MCS-T2A-copGFP-MSCV (CD523A-1) vector. This vector was mutated using the QuickChange site-directed mutagenesis kit II (Stratagene, Santa Clara, CA, USA). The lentiviral packaging vectors-psPAX2 and pVSVG were purchased from Addgene (Cambridge, MA, USA). The psPAX2 plasmids (2 µg), the expression vectors (2 µg) and the pVSVG plasmids (2 µg) were cotransfected into 293T cells, and the virus supernatants were collected after 48 h (2,000 rpm/min, 3 min). RAW264.7 cells were transduced with unconcentrated virus supernatant overnight in the presence of 8 mg/ml polybrene and selected in puromycin (0.5 mg/ml). The expression of VSIG4 and PDK2 was measured by western blot.

**Pdk2 and Stat3 shRNA silencing.** Lentiviral constructs with shRNAs directed against mouse Pdk2 and Stat3 in the pGV112 vectors, which were prepared by GeneChem (Shanghai, China). Primers are presented in Supplementary Table 4. Lentivirus was prepared by transient transfection of 293T cells with transfer vectors along with third-generation packaging constructs (pHelper 1.0 and pHelper 2.0). The viral titers were determined with serial dilution of virus-containing media on NIH3T3 cells. RAW264.7 cells were transfected with unconcentrated virus supernatant overnight in the presence of 8 mg/ml polybrene and selected in puromycin (0.5 mg/ml).

**Chromatin immunoprecipitation and qPCR.** BMDMs underwent cross-linking for 10 min with 1% formaldehyde in medium. Chromatin fragments were prepared, followed by immunoprecipitation with anti-Dnmt3a (#ab2850, Abcam, 1:1000, Rabbit), anti-STAT3 (#ab119352, Abcam, 1:1000, mouse) or rabbit/mouse IgG isotype mAbs and coupled to Dynabeads Protein G. DNA was phenol/chloroform extracted and ethanol precipitated in the presence of glycogen. DNA was purified with a PCR purification kit (#28104, Qiagen). qPCR was performed with special primers (Supplementary Table 5) flanking the putative Dnmt3a- and STAT3-binding sites. The input DNA was an aliquot of sheared chromatin before immunoprecipitation and was used for normalization of the samples to the amount of chromatin added to each ChIP.

**Vsig4 gene promoter methylation analyses.** The Sequenom MassARRAY platform (CapitalBio, Beijing, China) was used to perform the quantitative methylation analysis of Vsig4. This system uses matrix-assisted laser desorption/ionization time-of-flight (MALDI-TOF) mass spectrometry in combination with RNA base-specific cleavage (MassCLEAVE). A detectable pattern is then analyzed for its methylation status. PCR primers were designed with Methprimer (http://epidesigner.com) and were showed in (Supplementary Table 6). The spectra methylation ratios were generated by Epityper software version 1.0 (Sequenom, San Diego, CA).

**Vsig4 gene promoter luciferase reporter assay.** A840bp (-840/+1) fragment of Vsig4 gene promoter was cloned into the pGL3-basic expression vector using primers described in Supplementary Table 7. The pGL3-basic and the -840/+1 Vsig4 promoter constructs were fully methylated in vitro by M.SssI following the manufacturer's instructions (New England Biolabs (NEB), Beijing, China). The completeness of methylation was checked by measuring the extent of protection from digestion by the restriction enzymes BstUI (NEB). RAW264.7 cells were cultured in 12-well plates and were transfected with these plasmids by using amaxanucleofector (4D-Nucleofector, Lonza, Allendale, NJ, USA). At 48 h after transfection, cells were further treated with proinflammatory stimulation for an additional 12 h. Thereafter, cells were washed with PBS and lysed in Reporter Lysis Buffer (Invitrogen). Luciferase reporter activities were measured in triplicate using the Dual-Luciferase reporter assay system (Promega, Madison, WI, USA) according to the manufacturer's protocol, and quantified using the GloMax 96-well plate luminometer (Promega). The firefly luciferase to Renilla luciferase ratios were determined and were defined as the relative luciferase activity. Results are shown as the mean ± s.e.m. of a representative experiment performed in triplicate. To examine the transfection efficiency, RAW264.7 cells were transfected with a pmaxGFP control vector and transfection efficiency was assessed by in situ GFP expression according to manufacturer's protocol (Lonza).

**Statistical analysis.** Survival data from in vivo experiments were analyzed by a log-rank test performed on curves generated by GraphPad Prism 4.03 Software (MacKiev). For all other analysis, two-tailed, unpaired Student's t-tests with a 95% confidence interval performed on graphs generated in GraphPad Prism were used.

$p < 0.05$ was considered a statistically significant difference. All results shown are representative of at least three separate experiments.

**Data availability.** The authors declare that all data supporting the findings of this study are available within the article and its Supplementary Information Files or from the corresponding author upon request.

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

## Acknowledgements

Dr. Bin Li (Institute Pasteur of Shanghai, Chinese Academy of Sciences) gave invaluable suggestions. This work was supported by the General Program of National Natural Science Foundation of China (NSFC, No. 81361120400, 81222023 and 81771691), and The National Key Research and Development Program of China (2016YFA0502204).

## Author contribution

Y.C. and Y.W. designed the research and analyzed the data. J.L., S.G., and X.H. performed most experiments. Q.N. and W.Y. prepared MHV-3 virus. The manuscript was prepared by L.Z., and Y.C. and Z.F. conducted the plasmid mutation. B.D. and C.Y. conducted HFD-induced obesity.

## Additional information

**Competing interests:** The authors declare no competing financial interests.

