## [Peer review file · Nature Communications]

Reviewers' comments:

Reviewer #1 (Remarks to the Author):

This paper uncovers a function for VSIG4 in macrophages revealing it to be a regulator of mitochondrial pyruvate metabolism by controlling the expression of PDK2. There are interesting elements here but there are a lot of issues that need to be addressed to provide further important information here.

1. We need a lot more information on how VSIG4 regulates PDK2. What is the natural ligand for VSIG4 and under what circumstances will it regulate PDK2?
2. If VSIG4 is important for macrophage polarisation, how is it regulated itself? Does LPS repress it and if so how?
3. The paper concerns pyruvate metabolism and yet we have no analysis of pyruvate or the Krebs cycle. We need metabolomics here to assess key metabolites being considered. Also we need more information on how precisely mitochondrial ROS is being regulated here - there are connections in the literature but we need more experiments on the mechanism of ROS generation.
4. Finally how does the PI3K pathway regulate PDK2? We need more on the molecular basis for this pathway engaging with PDK2.

Reviewer #2 (Remarks to the Author):

This is very nice paper showing the possible role of VSIG4 in relation to obesity and insulin resistance as well as its importance in protection against viral infections. The authors also show novel mechanisms how VSIG4 provide intracellular signals extending previous findings in this field.

Major

The major findings are based on comparison of VSIG4^{-/-} with C57BL/6 WT controls. It is uncertain for this reviewer whether these mice are carefully matched and share the same back up genetics and back crossed properly to avoid unpredictable confounders.

If we assume that the results are solid and reliable a way to increase the cross over interest of this paper is to take the issue of complement C3 into perspective, which has been almost completely ignored since VSIG4 after all is regarded as a C3b, iC3b, C3c receptor with a regulatory effect on the alternative complement pathway convertase. I think the authors should either provide indications that this could be a possibility or rebut the possibility that complement may trigger the signals. C3 is produced locally in the macrophages which could be a substantial autocrine signal via VSIG4, which could easily be provided by using different knock down approaches in vitro. at least human C3 can also be purchased as can C3b.

The alternative protein name of VSIG4 (CRIg) should also be stated in the title and in the abstract to let all readers aware of which protein we are discussing.

Reviewer #1 (Remarks to the Author):

This paper uncovers a function for VSIG4 in macrophages revealing it to be a regulator of mitochondrial pyruvate metabolism by controlling the expression of PDK2. There are interesting elements here but there are a lot of issues that need to be addressed to provide further important information here.

Q1. We need a lot more information on how VSIG4 regulates PDK2. What is the natural ligand for VSIG4 and under what circumstances will it regulate PDK2?

A1: To address the underlying mechanisms for how VSIG4 regulate PDK2 activity, we carried out new experiments. We first found that knockdown *Stat3* (signal transduction and activator of transcription-3) expression in RAW 264.7 cells leads to downregulation of PDK2 (Fig. 6h). Secondly, we identified two STAT-3 binding sites (-1298 bp and -2934 bp) in *Pdk2* promoter region and chromatin immunoprecipitation (CHIP)-qPCR experiments reveal LPS stimulation promotes nuclear binding of pSTAT to the -1298-bp but not -2934-bp site of *Pdk2* promoter region. Additionally, the presence of VSIG4 signaling promotes this recruitment (Fig. 6i). These new data, together with previous results shown in Fig.6 c~g, clearly demonstrate that VSIG4 mediates PDK2 expression through the PI3K/Akt-STAT3 signaling pathway.

Previous work has shown that the complement C3b or iC3b appears to be the natural ligand of VSIG4. To address whether VSIG4 regulated macrophage activation and PDK2 expression are mediated by C3b or iC3b, we over-expressed *Vsig4* in BMDMs derived from complement *C3*^{-/-} mice and found that the LPS-induced the production of cytokines like IL-6 and IL-1β was still dramatically reduced in the absence of C3 (**Fig. 3h**). Next, we transfected the human monocytes line-THP-1 cells for over expressing human *Vsig4*, and these cells were induced to be macrophages by PMA stimulation. These cells were further stimulated by microbead-conjugated C3b, the results showed that microbead-C3b does not affect the basal and LPS-induced PDK2 expression (**Fig. 6j**). Moreover, overexpress *Vsig4* in *C3*^{-/-}BMDMs still increased basal and LPS-induced PDK2 expression (**Fig. 6j**). These combined data suggest that VSIG4 mediated PDK2 expression and cytokine production in macrophages is likely C3b independent. We recognize the importance of identification of VSIG4 ligand(s) and would like to commit our efforts and resources in a separate research project.

Q2. If VSIG4 is important for macrophage polarization, how is it regulated itself? Does LPS repress it and if so how?

A2: We appreciate the reviewer's insightful thought. We added more experiments to address the concern. Indeed, we found that the expression of *Vsig4* was down-regulated during inflammatory macrophage activation. PEMs and liver tissues that isolated from MHV-3 infected WT mice showed lower VSIG4 expression, as compared to uninfected littermates (**Fig. 7a and b**). Additionally, *in vitro* administration of proinflammatory mediators including LPS apparently can induce a quick transient decline of *Vsig4* gene and protein levels in PEMs (**Fig. 7c and d**), indicating that rapid *Vsig4* down-regulation is a common response of macrophage toward inflammatory stimulations.

We further analyzed the genomic DNA sequences of isolated PEMs that were treated with various proinflammatory cytokines. Interestingly, MALP-2, IFN- γ , PolyI:C and LPS are all capable of inducing very high incidence (98 ~ 100%) of methylation at a special CpG site (-372 bp) within the promoter region of *Vsig4* gene, which is significantly elevated from a 75% basal methylation at this site in the untreated cells (**Fig. 7e**). Interestingly, ChIP-qPCR assays reveal a significant enrichment of mammalian DNA methyltransferases-3a (Dnmt3a) binding to the regulatory elements proximal to the starting site of *Vsig4* transcription (**Fig. 7f**). In concordance, blocking Dnmt3a activity by an inhibitor, 5-aza-2'-deoxycytidine (AZAdC, 10 μ M), could effectively restore VSIG4 expression in PEMs under inflammatory conditions (**Fig. 7g**). These combined data demonstrate that Dnmt3a represses *Vsig4* expression through inducing DNA methylation under inflammatory conditions (including LPS administration).

Q3. The paper concerns pyruvate metabolism and yet we have no analysis of pyruvate or the Krebs cycle. We need metabolomics here to assess key metabolites being considered. Also we need more information on how precisely mitochondrial ROS is being regulated here - there are connections in the literature but we need more experiments on the mechanism of ROS generation.

A3: The reviewer raises important points. Actually, we think VSIG4 affects pyruvate metabolism due to our data showed that the levels of Pyruvate and Acetyl-CoA in LPS activated *Vsig4*⁺RAW264.7 cells were dramatically lower than their control counterparts (**Fig. 4a**). Additionally, VSIG4 signaling appeared to drastically down-regulate oxygen consumption in RAW264.7 cells after 6h of LPS exposure, both in basal and maximal OCR (**Fig. 4b and c**). These combined data imply that VSIG4 inhibits mitochondrial oxidation during macrophage activation. Moreover, the levels of ATP, which is majorly derived from Krebs cycle, was markedly higher in basal and LPS-treated *Vsig4*⁺RAW264.7 cells than the

control counterparts, suggesting the control cells (M1 phenotype) need higher energy for their biofunction (**as following**). We also detected the concentration of TCA metabolites including malate, citrate, AKG and succinate in *Vsig4*⁺ RAW264.7 cells and their control counterparts, and the results showed that the presence of VSIG4 signaling slightly but not significantly decreased the concentrations of these molecules (**data not shown**). Therefore, although the exact mechanism how the secretion of mtROS was affected by Krebs cycle, the reduction in release of mtROS from macrophages might be a result of negative regulatory role of VSIG4 in controlling metabolic rate of Krebs cycle. However, we have not proper place to present these data but discussed in the revised manuscript.

Figure 1 VSIG4 signaling affects the concentration of ATP in basal and LPS-activated RAW264.7 cells.

Q4. Finally how does the PI3K pathway regulate PDK2? We need more on the molecular basis for this pathway engaging with PDK2.

A4: As described in response to the reviewer's Q1, we carried out new experiments to address the roles of PI3K/AKT in signaling STAT3 phosphorylation and the subsequent p-STAT3 dependent induction of PDK2 expression. Our new data suggest that knockdown *Stat3* expression in RAW 264.7 cells leads to downregulation of PDK2 (**Fig. 6h**). This is likely due to the binding of pSTAT3 to the STAT-3 binding sites at -1298 bp of the *Pdk2* promoter region and as a result, promoting expression of the gene (**new Fig.6i**). In addition, our new CHIP-qPCR data indicate that VSIG4 signaling appears to promote LPS-induced recruitment of pSTAT-3 at -1298 bp site of *Pdk2* promoter region (**Fig. 6i**). Along with previous presented data in Fig.6 c~g, we clearly demonstrate that VSIG4-PI3K/Akt-STAT3 axis can promote transcriptional *Pdk2* expression.

Reviewer #2 (Remarks to the Author):

This is very nice paper showing the possible role of VSIG4 in relation to obesity and insulin resistance as well as its importance in protection against viral infections. The authors also show novel mechanisms how VSIG4 provide intracellular signals extending previous findings in this field.

Major

Q1 : The major findings are based on comparison of VSIG4^{-/-} with C57BL/6 WT controls. It is uncertain for this reviewer whether these mice are carefully matched and share the same back up genetics and back crossed properly to avoid unpredictable confounders.

A1: We appreciate the reviewer's careful reading and critical concerns. In the revised manuscript, we described our animals as the following: The complement C3-deficient (C3^{-/-}) mice (#003641) and the C57BL/6 (B6) mice were purchased from Jackson Laboratory. The *Vsig4*- deficient (*Vsig4*^{-/-}) mice were kindly provided by Dr. M. van Lookeren Campagne (Department of Immunology, Genentech, CA, USA). The *Pdk2*^{-/-} mice were provided by Dr. C.R. Harris (Rutgers Cancer Institute of New Jersey, USA). All mice were backcrossed 10 times onto the B6 background to avoid unpredictable confounders. Specific pathogen-free male and age-matched mice (8~12 weeks old) were used for all experiments.

Q2: If we assume that the results are solid and reliable a way to increase the cross over interest of this paper is to take the issue of complement C3 into perspective, which has been almost completely ignored since VSIG4 after all is regarded as a C3b, iC3b, C3c receptor with a regulatory effect on the alternative complement pathway convertase. I think the authors should either provide indications that this could be a possibility or rebut the possibility that complement may trigger the signals. C3 is produced locally in the macrophages which could be a substantial autocrine signal via VSIG4, which could easily be provided by using different knock down approaches in vitro. at least human C3 can also be purchased as can C3b.

A2: We totally agree with the reviewer's speculations and realized it's imperative for us to address the question in this manuscript. To resolve this issue, we added more

experiments. Indeed previous work has shown that the complement C3b or iC3b is the natural ligand of VSIG4. To address whether VSIG4 regulated macrophage activation and PDK2 expression are mediated by C3b or iC3b, we over-expressed *Vsig4* in BMDMs derived from complement *C3*^{-/-} mice and found that the LPS-induced the production of cytokines like IL-6 and IL-1 β was still dramatically reduced in the absence of C3 (**Fig. 3h**). Next, we transfected the human monocytes line-THP-1 cells for over expressing human *Vsig4*, and these cells were induced to be macrophages by PMA stimulation. These cells were further stimulated by microbead-conjugated C3b, the results showed that microbead-C3b does not affect the basal and LPS-induced PDK2 expression (**Fig. 6j**). Moreover, overexpress *Vsig4* in *C3*^{-/-}BMDMs still increased basal and LPS-induced PDK2 expression (**Fig. 6j**). These combined data suggest that VSIG4 mediated PDK2 expression and cytokine production in macrophages is likely C3b independent.

Q3: The alternative protein name of VSIG4 (CRIg) should also be stated in the title and in the abstract to let all readers aware of which protein we are discussing.

A3: We thank the reviewer's comments and suggestions. The title has been revised as: VSIG4 (CRIg) Negatively Regulates Proinflammatory Macrophage Activation *via* Reprogramming Mitochondrial Pyruvate Metabolism. Moreover, in the abstract, the sentence " We here illustrate that V-set immunoglobulin-domain-containing 4 (VSIG4, also called complement receptor of the immunoglobulin superfamily, CRIg), a protein specifically expressed in resting macrophages, inhibits macrophage activation in response to lipopolysaccharide (LPS) exposure in vitro." was added.

Reviewers' comments:

Reviewer #1 (Remarks to the Author):

The authors have adequately addressed my concerns.

Reviewer #2 (Remarks to the Author):

I have read through the manuscript carefully and they have addressed my questions very well and also the other reviewer I think. Thus from my point of view I think it is acceptable.

Reviewer #3 (Remarks to the Author):

The authors have added DNA methylation performed by massarray sequenome which is solid. The question however is whether the increase in DNAm level is biologically relevant. Have the authors shown relevance of position -372 for gene expression of Vsig4. This needs to be addressed.

I would further suggest that the authors should show the DNA methylation by sequenome following treatment (one or more) and show the quantitation for all CpG's in the region of the promoter

The DNMT chip-qPCR is hard to judge. The recoveries are very low, no positive or negative controls were added.

The big question is again what the relevance is of the chip-qPCR? Does the 25% increase in DNAm result in increased DNMT3a occupancy which in turn leads reduced expression. They do not show DNMT3a following treatment. Second, as before, no positive or negative control so it is impossible to judge the relevance of the chip-seq.

Reviewers' comments:

Reviewer #1 (Remarks to the Author):

The authors have adequately addressed my concerns.

Reviewer #2 (Remarks to the Author):

I have read through the manuscript carefully and they have addressed my questions very well and also the other reviewer I think. Thus from my point o view I think it is acceptable.

Reviewer #3 (Remarks to the Author):

Q1: The authors have added DNA methylation performed by massarray sequenome which is solid. The question however is whether the increase in DNAm level is biologically relevant. Have the authors shown relevance of position –372 for gene expression of *Vsig4*. This needs to be addressed.

A1: The reviewer raises some important points. To confirm that the expression of *Vsig4* is controlled by promoter methylation in mouse BMDMs, we used 5-aza-2'-deoxycytidine (AZAdC), a specific inhibitor of DNA methyltransferases, to show that inhibition of DNA methylation can effectively restore VSIG4 expression under inflammatory conditions (**Figure. 7f**). These combined data suggest that the expression of *Vsig4* is regulated by DNA methylation. And in regard to the issue, we are carrying forward a separate project by collecting clinical samples, including hepatocellular carcinomas, tissues from HBV-related acute-on-chronic liver failure (HBV-ALCF) and non-small-cell lung cancers, for analyses of the biological relevance of *Vsig4* DNAm with the host immunity. Nevertheless, with the results expected to come out in the next year, we consider it is beyond the scope of our current manuscript.

Q2: I would further suggest that the authors should show the DNA methylation by sequenome following treatment (one of more) and show the quantitation for all CpG's in the region of the promoter.

A2: We thank the reviewer's comments and suggestions. In the revised manuscript, we have showed the quantitation for all CpG's around the *Vsig4* promoter region in **Supplementary Table 1**.

Q3: The DNMT chip-qPCR is hard to judge. The recoveries are very low, no positive or negative controls were added.

A3: We appreciate the reviewer's careful reading and critics. In the revised manuscript, BMDMs that treated with PBS were used as negative control, and Chip-qPCR showed that dramatically higher levels of Dnmt3a recruitment to *Vsig4* promoter after LPS or MALP-2 administration as compared to PBS-treated counterparts, with results were showed in **Fig. 7h**. We also improved our Chip-qPCR working system and now the recoveries are 10 times higher than previous data at least (**Fig. 7h**).

Q4: The big question is again what the relevance is of the chip-qPCR? Does the 25% increase in DNAm result in increased DNMT3a occupancy which in turn leads reduced expression. They do not show DNMT3a following treatment. Second, as before, no positive or negative control so it is impossible to judge the relevance of the chip-seq.

A4: We totally agree with the reviewer's speculations and realized it's imperative for us to address the question in this manuscript. In the revised manuscript, we showed that the expression of Dnmt3a was upregulated in BMDMs that treated with various kinds of proinflammatory cytokines, nevertheless, the expression of Dnmt1 and Dnmt3b was very low and not affected under such conditions (**Fig. 7g**). Therefore, we think that Dnmt3a is the special DNA methyltransferase that promotes position -372 DNA methylation (25% increase), which leads to reduction in VSIG4 expression. Together with data showed in **Fig. 7f** that the Dnmts inhibitor, 5-aza-2'-deoxycytidine (AZAdC) was able to effectively restore VSIG4 expression under inflammatory conditions, we demonstrate that Dnmt3a controls VSIG4 gene repression through fast methylation of the transcriptional initiation site.

Reviewers' comments:

Reviewer #3 (Remarks to the Author):

The authors have addressed some technical questions regarding DNA methylation. Nevertheless, the statement that DNA methylation at a single CpG site is the mechanism responsible for the downregulation of VSIG4 in inflammation is a (too) strong one, and therefore requires further exploration.

1. The authors show that DNMT3a binds to the VSIG4 promoter after LPS treatment (qPCR - Fig 7h). They also claim that levels of DNMT3a are increased after stimulation (western blot - Fig 7g). While this is true for some stimuli, the levels do not appear to increase after LPS treatment. Does this indicate that higher levels of DNMT3a protein is not required to silence VSIG4 after LPS treatment?
2. DNA methylation analysis of the VSIG4 promoter after exposure to TNF, IL6 and IL1b is missing in Figure 7e
3. The same stimuli should be shown in all Figure 7 e, f, g, and h.
4. The authors use 5-aza to show that demethylation of the VSIG4 promoter leads to higher expression. However 5-aza leads to demethylation globally, which does not address the question of whether the -374 site is important. The required experiment would be a promoter reporter construct where HhaI, HpaI and SssI are used to specifically methylate CpG sites within the VSIG4 promoter, followed by stimulation. This would then indicate if the -374 CpG site is the one controlling gene expression.
5. The authors only show one time-point in Figure 7: 12 hours. In order to resolve the order of events that lead to lower protein levels of VSIG4 after stimulation (i.e. does DNA methylation occur first and is subsequently silencing the gene?), the authors should perform DNA methylation and gene expression analysis on earlier time-points to see which changes first.
6. Minor: In table S1 the CpG site is called '-374' and in Figure 7 it is '-372'.

Reviewer #3 (Remarks to the Author):

The authors have addressed some technical questions regarding DNA methylation. Nevertheless, the statement that DNA methylation at a single CpG site is the mechanism responsible for the downregulation of VSIG4 in inflammation is a (too) strong one, and therefore requires further exploration.

Q: 1. The authors show that DNMT3a binds to the VSIG4 promoter after LPS treatment (qPCR - Fig 7h). They also claim that levels of DNMT3a are increased after stimulation (western blot - Fig 7g). While this is true for some stimuli, the levels do not appear to increase after LPS treatment. Does this indicate that higher levels of DNMT3a protein is not required to silence VSIG4 after LPS treatment?

A1: We appreciate the reviewer's careful reading and critical concerns. To address the issue, we ordered a new vial of LPS from Sigma and conducted multiple experiments in measuring LPS response. Our results demonstrated that Dnmt3a expression is indeed upregulated in response to LPS. Fig. 7e is one representative results of 4 experiments, with the results of other three repeats shown as the following (Fig. 1).

Fig.1 Enhancing Dnmt3a expression by proinflammatory stimuli. BMDMs were treated with various proinflammatory stimuli, and the expression of Mnmt3a was measured by western-blot at 12h. Three separate experimental results are shown.

Q2. DNA methylation analysis of the VSIG4 promoter after exposure to TNF, IL-6 and IL1b is missing in Figure 7e.

A2: To address the concern, we carried new experiments as suggested by the reviewer, using in match with the individual stimulations presented in Figure 7d, for assessment of corresponding methylation on the *Vsig4* promoter. The results are presented in the revised supplementary Fig. 10b.

Q3. The same stimuli should be shown in all Figure 7 e, f, g, and h.

A3: We agree with the reviewer's comments and have accordingly revised Figure 7e, f and supplementary Fig. 10 b and c.

Q4. The authors use 5-aza to show that demethylation of the VSIG4 promoter leads to higher expression. However 5-aza leads to demethylation globally, which does not address the question of whether the -374 site is important. The required experiment would be a promoter reporter construct where HhaI, HpaI and SssI are used to specifically methylate CpG sites within the VSIG4 promoter, followed by stimulation. This would then indicate if the -374 CpG site is the one controlling gene expression.

A4: The reviewer raises an important point here. To address the issue, we demonstrated that 5-aza (AZAdC), a general methyltransferase inhibitor, prevents *Dnmt3a* upregulation in BMDMs in response to *in vitro* proinflammatory stimuli, and by thus it augments the expression of VSIG4. This indicates that 5-aza counteracts with proinflammatory mediator-stimulated *Vsig4* downregulation (Fig.7f).

From the literature (Kumar P, *et al.* Biosci Rep. 2009 Feb;29(1):57-70), we know two motifs of CpG island, one that can be recognized by HhaI methyltransferase is “gcgc” , and the other by HpaII is “ccgg”. By analyzing the sequence of the *Vsig4* promoter between positions -2000bp and +1 relative to the TSS, we have realized that there is no any recognition sequence in this area that could be methylated by HhaI and HpaII methyltransferase. Therefore, we have assumed that the *Vsig4* promoter activity is not affected by HhaI and HpaII methyltransferase. On the other hand, the recognition sequence for SssI methyltransferase is “cg” , and there are many “cg” motifs in the 2000bp of *Vsig4* promoter region, which limited further assessments.

To validate that the methylation in *Vsig4* promoter region by Dnmt3a is critical for controlling *Vsig4* expression, we tried an alternative approach, by cloning a fragment of *Vsig4* promoter (-860/+1) into PLG3-Basic luciferase reporter vectors and measuring the promoter activity. By transfecting both 293T cells and RAW264.7 cells, we show that the basal levels of *Vsig4* promoter-driven luciferase activity are lower than PLG3-Basic counterparts, especially in 293T cells (Fig.2), suggesting there might be silencing motif(s) in the 840bp of *Vsig4* promoter region that attenuate *Vsig4*- promoter activity. This result is very interesting yet it becomes a challenge for identification of the repressing factors for *Vsig4* promoter activity. Nevertheless, we believe further elucidation of the underlying details is beyond the scope of our current report.

Fig.2 Detection of *Vsig4* promoter activity. Luciferase activity of the *in vitro* pGL3-Basic vector, the *Vsig4* promoter construct (-840/+1) and positive control (pGL3-Basic with a CMV promoter) transfected into 293T and RAW264.7 cells. One set representative data of three independent experiments is shown.

To examine the effect of promoter methylation on *Vsig4* promoter activity, RAW264.7 cells were transfected *in vitro* with SssI methylated pGL3-Basic and -840/+1 *Vsig4* promoter constructs, after 48h, cells were further stimulated by proinflammatory stimuli for an additionally 12h. Compared to un-methylated controls, the *Vsig4* promoter activity was reduced dramatically after methylated by SssI M (Fig. 7f). However, proinflammatory stimuli are incapable of further reducing the luciferase activity in SssI methylated -820/+1 *Vsig4* promoter constructs (Fig. 7f). Together these data demonstrate that DNA methylation really controls *Vsig4* expression.

We found that a CpG at -374bp site in *Vsig4* promoter region was methylated by

proinflammatory stimuli as detected by using the Sequenom MassARRAY platform (Supplementary Fig. 10b). Furthermore, ChIP-qPCR assays reveal a significant enrichment of Dnmt3a in -374bp of *Vsig4* promoter region after the BMDMs were treated with proinflammatory stimuli (Supplementary Fig. 10c). However, since only 825bp of *Vsig4* promoter region was detected by the Sequenom MassARRAY platform, we do not have sufficient evidence to exclude that other CpG islands in the *Vsig4* promoter region might also be methylated by Dnmt3a and involve in controlling *Vsig4* gene transcription. For this reason, we transferred the related data to the supplementary Fig.10. We hope the reviewer and editor accept our position, as it stands with current data and meanwhile leaves the uncertainty for future clarification.

Q5. The authors only show one time-point in Figure 7: 12 hours. In order to resolve the order of events that lead to lower protein levels of VSIG4 after stimulation (i.e. does DNA methylation occur first and is subsequently silencing the gene?), the authors should perform DNA methylation and gene expression analysis on earlier time-points to see which changes first.

A5: We appreciate the reviewer's insightful thought. We compared the expression of Dnmt3a and VSIG4 at earlier time of proinflammatory cytokine treatment. For example, at 3h of proinflammatory cytokine administration, the protein expression of Dnmt3a was very low, appearing to be not affected by proinflammatory cytokine treatment. However, the protein level of Dnmt3 was increased markedly at 6h of LPS, TNF- α , MALP-2 and Poly I:C treatments, whereas the expression of VSIG4 was not downregulated dramatically at this time point (Fig. 3a). At the same time, the *Vsig4* gene transcription was not affected at the same time (Fig. 3b). On the other hand, the methylation levels at CpG at -374bp site and -113/-118bp sites were increased at 6h post some stimulations (Fig. 3c). Together these data suggest that DNA methylation occurs prior to repression of *Vsig4* expression.

To make our manuscript coherent and concise, we prefer not to present these data in the current already lengthy manuscript, rather to show them to the reviewers only in our rebuttal letter here. We hope the reviewer and editor accept our considerations.

Fig.3 Enhancing Dnmt3a expression and *Vsig4* promoter methylation at 6h of proinflammatory stimuli. BMDMs were treated with proinflammatory mediators for 3h and 6h, respectively. (a) Expression of Dnmt3a and VSIG4 was measured by western-blot. (b) BMDMs were treated with proinflammatory mediators for 6h, and *Vsig4* transcription detected by RT-qPCR. NS: not significant different. (c) BMDMs were treated with proinflammatory mediators for 6h, the methylation levels of CpG sites in *Vsig4* promoter regions were subjected to assessment through the Sequenom MassARRAY platform. Results of quantitative methylation analysis are shown in a different color scale, TSS, transcriptional start site.

Q6. Minor: In table S1 the CpG site is called ‘-374’ and in Figure 7 it is ‘-372’.

A6: Thanks the reviewer’s pick, we have made the correction and the CpG site is -374 bp.

REVIEWERS' COMMENTS:

Reviewer #3 (Remarks to the Author):

The authors have done a good job at addressing my comments. This includes several additional experiments that have strengthened the study.